# Topological acoustic triple point

Sungjoon Park[1,2,3,4], Yoonseok Hwang [1,2,3,4], Hong Chul Choi[1,2] & Bohm-Jung Yang [1,2,3 ✉]

Acoustic phonon is a classic example of triple degeneracy point in band structure. This triple point always appears in phonon spectrum because of the Nambu–Goldstone theorem. Here, we show that this triple point can carry a topological charge $\mathfrak{q}$ that is a property of three-band systems with space-time-inversion symmetry. The charge $\mathfrak{q}$ can equivalently be characterized by the skyrmion number of the longitudinal mode, or by the Euler number of the transverse modes. We call triple points with nontrivial $\mathfrak{q}$ the topological acoustic triple point (TATP). TATP can also appear at high-symmetry momenta in phonon and spinless electron spectrums when $O_h$ or $T_h$ groups protect it. The charge $\mathfrak{q}$ constrains the nodal structure and wavefunction texture around TATP, and can induce anomalous thermal transport of phonons and orbital Hall effect of electrons. Gapless points protected by the Nambu–Goldstone theorem form a new platform to study the topology of band degeneracies.

[1] Center for Correlated Electron Systems, Institute for Basic Science, Seoul 08826, Korea. [2] Department of Physics and Astronomy, Seoul National University, Seoul 08826, Korea. [3] Center for Theoretical Physics (CTP), Seoul National University, Seoul 08826, Korea. [4]These authors contributed equally: Sungjoon Park, Yoonseok Hwang. ✉email: bjyang@snu.ac.kr

Classification of topological phases of matters has been a topic of intensive research[1–3]. An important conclusion drawn from these investigations is that gap closing points in the band structure are often characterized by a topological charge. A famous example is the Weyl point, whose gaplessness is protected by the Chern number[4]. However, in nature, there is a different class of gap closing points that are enforced by the Nambu–Goldstone (NG) theorem, whose topological characteristics have been largely unexplored yet[5–7]. A familiar example is the acoustic phonons, which are NG bosons resulting from breaking the translational symmetries, and exist even in classical systems. Because three translational symmetries are broken, there are three gapless acoustic phonons forming a triple point at the Brillouin zone (BZ) center, which we refer to as the acoustic triple point (ATP).

Here, we show that an ATP in elastic crystals with time-reversal symmetry can carry a topological charge q. q is a topological charge that can be defined for a real-symmetric three-band Hamiltonian, and consists of a pair of well-known topological charges: the skyrmion number $\mathfrak{n}_{sk}$ and the Euler number $\mathfrak{e}$. Hence, an ATP with nontrivial q is dubbed the "topological ATP" (TATP). The topological charge q is strictly defined only when the total number of energy bands is fixed to three, so that it falls under the recently proposed category of "delicate" (topological) charge[8], which is distinct from the stable charge[9,10] such as the Chern number or the fragile charge[11–17] such as the Euler number[18]. In general, the delicate charge is defined for small number of bands and thus it is not well defined in electronic system, where the total number of electron bands easily exceeds the relevant number. In contrast, the number of phonon energy bands is fixed by the number of atoms in the unit cell, so that there is a possibility that phonons can be exactly characterized by the delicate charge.

Although only phonons in monatomic lattices have precisely three bands, the theory of elasticity naturally yields an effective three-band description of the ATP, which are the three NG modes. In this sense, we find that TATP protected by the NG theorem is ubiquitous in elastic materials. Interestingly, the triple points with nontrivial q can also be symmetry-protected at high-symmetry momentum in $\mathcal{PT}$ symmetric elastic systems, and even in $\mathcal{PT}$ symmetric electronic systems with negligible spin-orbit coupling, where $\mathcal{P}$ and $\mathcal{T}$ are inversion and time-reversal symmetries, respectively. The TATP protected by the NG theorem has a linear dispersion, while the symmetry-protected triple point has a quadratic dispersion around the triple point. However, since both share the same topological charge, we refer to both types of triple points as the TATP.

A characteristic feature of both the linearly and quadratically dispersing TATPs is the energy gap between the highest energy band (L mode) and the two lower energy bands (T modes), except at the triple point, see Fig. 1a. This gap is necessary to define the topological charge q, and this feature distinguishes the TATPs from the triple points created by band inversion[19–26] and the spin-1 Weyl point[27–32], see Fig. 1b.

Because having a nontrivial q implies nontrivial $\mathfrak{n}_{sk}$ and $\mathfrak{e}$ for the longitudinal and the transverse modes, respectively, it has interesting consequences for the nodal structure. For example, there must be at least four nodal lines formed between the T modes emanating from the TATP. Also, because the nonzero q is accompanied by nontrivial winding texture of the wavefunctions around the TATPs, systems with TATPs can show anomalous transport of phonon angular momentum or electronic orbital.

## Results

**Topological charge**. Let $H_{\mathbf{k}}$ denote either the dynamical matrix of phonon or the Hamiltonian matrix of electron, and let $\epsilon_{\mathbf{k},n}$ and $\mathcal{E}_{\mathbf{k},n}$ be the eigenvector and eigenvalue of $H_{\mathbf{k}}$, respectively. Note

that when $H_{\mathbf{k}}$ is the dynamical matrix of phonon, $\mathcal{E}_{\mathbf{k},n} = \omega_{\mathbf{k},n}^2$ where $\omega_{\mathbf{k},n}$ is the phonon energy, and when $H_{\mathbf{k}}$ is the Hamiltonian matrix of electron, $\mathcal{E}_{\mathbf{k},n}$ is the electron energy. This means that insofar as the topology of ATP is concerned, there is no difference between the dynamical matrix of phonon and the Hamiltonian matrix of electron. Henceforward, we blur the difference between phonon and electron and refer to $H_{\mathbf{k}}$ as the Hamiltonian, and clarify the difference when a possibility of confusion arises.

To define q, we require that $H_{\mathbf{k}}$ be a $3 \times 3$ real-symmetric matrix. The three-band condition is satisfied by the phonons in a monatomic lattice, which have only three phonon bands. Even when there is more than one atom per unit cell, and therefore more than three phonon energy bands, this condition is satisfied near the ATP, which can be described by the elastic continuum Hamiltonian[33] (see Theory of elastic continuum in Methods and Supplementary Note 3). The condition that $H_{\mathbf{k}}$ be real is satisfied if there is $\mathcal{PT}$ symmetry that satisfies $(\mathcal{PT})^2 = 1$, in which case it is possible to choose $\mathcal{PT} = K$, where $K$ is the complex conjugation operator. It is useful to note that even if $\mathcal{P}$ is absent in the crystal, it is restored in the elastic continuum limit. Thus, the elastic continuum Hamiltonian in time-reversal symmetric crystals always satisfies the three-band and the reality conditions (see Theory of elastic continuum in Methods).

The dynamical matrix of the isotropic elastic continuum offers a simple picture of the main characteristics of q:

$$[H_{\mathbf{k}}]_{\alpha\beta} = v_T^2 k^2 \delta_{\alpha\beta} + (v_L^2 - v_T^2)k_\alpha k_\beta, \tag{1}$$

where $v_L$ and $v_T$ are the longitudinal and transverse velocities, respectively. Defining $k = \sqrt{k_x^2 + k_y^2 + k_z^2}$ and $\tilde{k} = \sqrt{k_x^2 + k_y^2}$, the eigenstates are $\boldsymbol{\epsilon}_{\mathbf{k},L} = \frac{1}{k}(k_x, k_y, k_z)$, $\boldsymbol{\epsilon}_{\mathbf{k},T_1} = \frac{1}{\tilde{k}}(-k_y, k_x, 0)$, $\boldsymbol{\epsilon}_{\mathbf{k},T_2} = \frac{1}{k\tilde{k}}(-k_x k_z, -k_y k_z, k_x^2 + k_y^2)$, whose eigenvalues are given by $v_L^2 k^2$, $v_T^2 k^2$, and $v_T^2 k^2$ respectively. To define q, we consider a sphere surrounding the triple point at k = 0. On this sphere, notice that the L mode has the skyrmion number $\mathfrak{n}_{sk} = 1$, see Fig. 1c. Furthermore, the T modes span the tangent space of the sphere in the momentum space, so that they have Euler number $\mathfrak{e} = 2$ as is well known. Alternatively, the Euler number can be computed by counting the winding number of the Wilson loop spectrum[34,35], see Fig. 1d (see also Computation of Euler number using Wilson loop in Methods). Therefore, we define the topological charge $q = (\mathfrak{n}_{sk}, \mathfrak{e})$, where it can be shown that the constraint $\mathfrak{e} = 2\mathfrak{n}_{sk}$ must be satisfied. This discussion can be generalized to any $3 \times 3$ real-symmetric Hamiltonian as long as there is a gap between the L and the T modes, see Homotopy description of q in Methods and Supplementary Note 1.

**Phonons in CsCl**. Although the topological charge q was defined for a three-band system, the topological charge is still meaningful in multiband systems. To demonstrate this, let us study the phonon spectrum of CsCl lattice, which has two atoms per unit cell (see Fig. 2a), and therefore six phonon bands. We show the phonon spectrum ($\omega_{\mathbf{k}}$) obtained from first-principles calculations (see Details of ab initio calculations in Methods) in Fig. 2b. Near Γ with $\mathbf{k} \neq 0$, we see that the two lowest acoustic modes are gapped from the others. Therefore, we can compute the Wilson loop spectrum for these two acoustic phonons, which we show in Fig. 2c. From the winding structure of the Wilson loop spectrum, we see that $|\mathfrak{e}| = 2$. It is important to note that although the CsCl lattice has six phonon bands, we can still define the Euler number for the T modes. This is because $\mathfrak{e}$ can be defined for any two bands that are isolated from the others by a gap, so that it is not sensitive to the total number of energy bands present in the system. In contrast, $\mathfrak{n}_{sk}$ and q are properties of a $3 \times 3$ Hamiltonian, so that they are not well defined here in a strict sense.

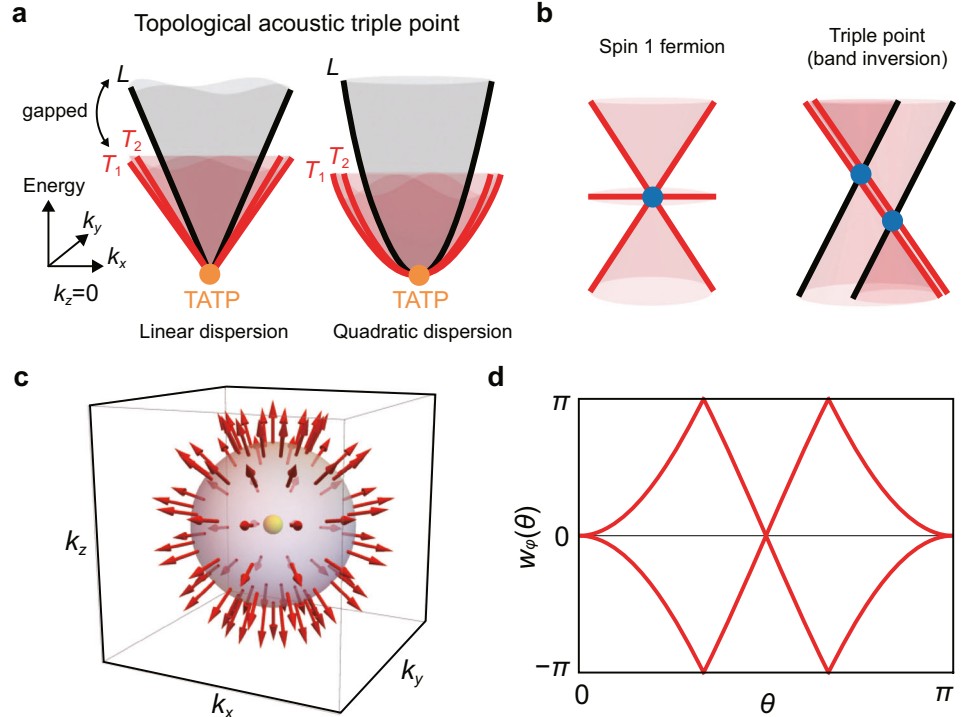

**Fig. 1 Topological acoustic triple point (TATP). a** TATP with $\mathfrak{q} \equiv (\mathfrak{n}_{sk}, \mathfrak{e}) = (1, 2)$ can appear with either a linear dispersion or a quadratic dispersion. Note that there is a gap between the $L$ and the $T$ modes away from the triply degenerate point. **b** TATP is distinct from spin-1 Weyl point, which is protected by Chern numbers. TATP is also distinct from the triple point formed by band inversion, where it is not possible to separate the highest energy band from two lower energy bands away from the triple point. **c** Skyrmion texture of longitudinal phonon on the sphere wrapping a TATP. The transverse modes span the tangent plane of the sphere. **d** Wilson loop spectrum for the transverse modes as a function of the polar angle $\theta$ computed on a sphere wrapping the TATP. $|\mathfrak{e}|$ is given by the number of times that one of the two branches of $w_\phi(\theta)$ crosses $\pi$.

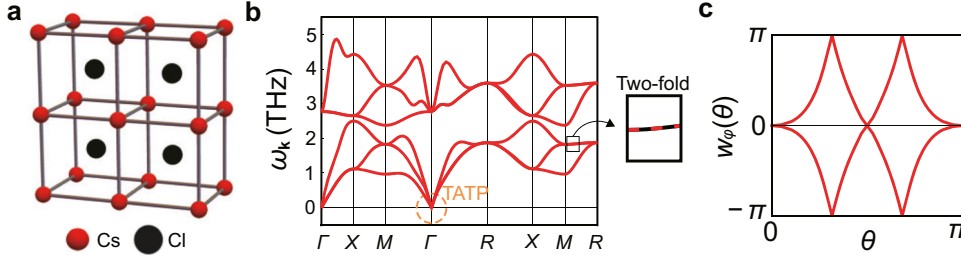

**Fig. 2 TATP in CsCl. a** CsCl lattice structure. **b** Phonon spectrum ($\omega_{\mathbf{k}}$) of CsCl along the high-symmetry lines obtained by first-principles calculations. Acoustic phonons at $\Gamma$ carry $\mathfrak{q} = (1, 2)$. $\mathfrak{q}$ cannot be defined for the triple degeneracy at the $R$ point, because lower two bands cannot be fully separated from the highest energy band owing to the degeneracy between upper two bands along the $RM$ direction. **c** Wilson loop spectrum of the two lowest acoustic phonons near the $\Gamma$ point.

Therefore, $\mathfrak{q} = (\mathfrak{n}_{sk}, \mathfrak{e})$ reduces to $\mathfrak{e}$ when the number of phonon bands is larger than 3. However, we can recover the topological charge $\mathfrak{q}$ by using the continuum theory or the Löwdin partitioning (see Continuum theory in Results).

We note that the triply degenerate optical modes at $\Gamma$ is also a TATP (see Details of ab initio calculations in Methods), whereas $\mathfrak{q}$ cannot be defined for the triple degeneracy at the $R$ point, because it is not possible to consistently separate one of the energy bands from the other two on a sphere surrounding the triple degeneracy (see Fig. 2b).

**Continuum theory**. Here, we discuss how the continuum theory constrained by the crystalline symmetries allows us to extend the discussion of TATP to general multiband systems. Let us first consider the gapless acoustic phonons, which are conventionally

described by the elastic continuum theory. This naturally yields a $3 \times 3$ elastic continuum Hamiltonian (dynamical matrix) description of the acoustic phonons, whose specific form is constrained by the 32 point group symmetries allowed by the crystal[36]. Because the triple point is always present due to the gaplessness of phonons, all 32 point group symmetries are meaningful, as long as the $\mathcal{T}$ symmetry is present (see Theory of elastic continuum in Methods).

For simplicity, let us focus on the elastic continuum Hamiltonian constrained by the cubic symmetries. Because we are interested in the topological properties, it is sufficient to examine only the traceless part of the Hamiltonian, which takes the form

$$H_{\mathbf{k}} = \sum_n f_n(\mathbf{k})\lambda_n, \tag{2}$$

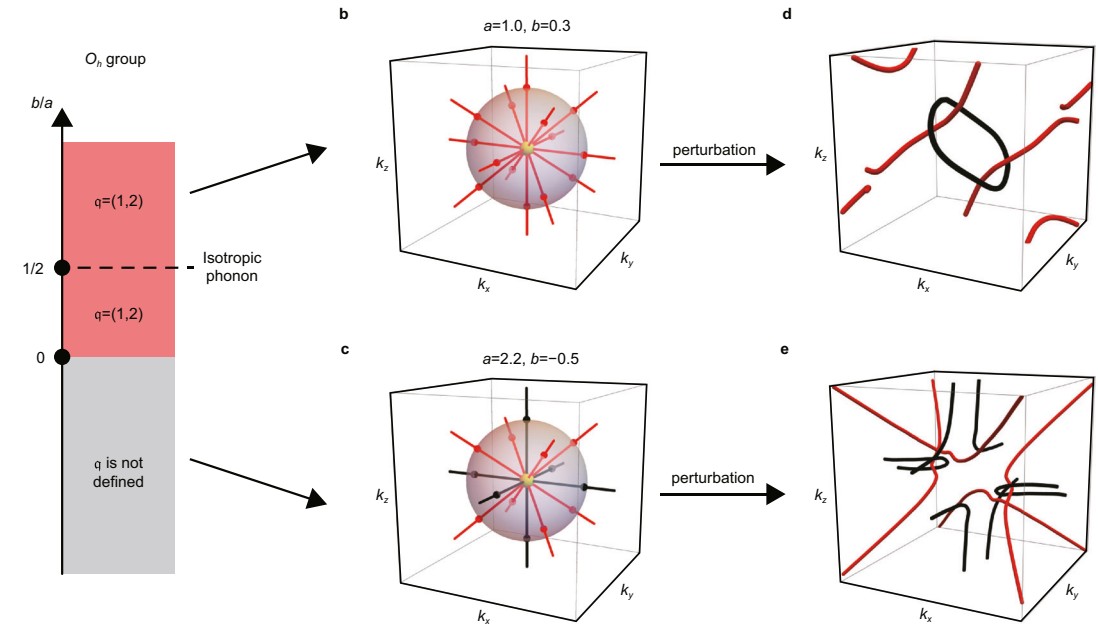

**Fig. 3 Acoustic triple points in cubic systems. a** Phase diagram for the elastic continuum Hamiltonian in Eq. (2) for acoustic phonons in cubic systems. **b, c** Nodal structure for $b/a > 0$ (**b**) and $b/a < 0$ (**c**). The black (red) lines are band degeneracies between the upper (lower) two bands. Notice that there are two types of nodal lines, one along the $k_x$, $k_y$, and $k_z$ axes and another along the lines that satisfy $|k_x| = |k_y| = |k_z|$. For $b/a > 0$, the band degeneracies occur only between the lower two bands. However, the eigenvalues of the degenerate bands along the $k_x$, $k_y$, and $k_z$ axes increase as $b/a$ decreases, so that when $b/a < 0$, these degeneracies occur between the upper two bands instead of the lower two bands. When we perturb the Hamiltonian in **b** such that the conditions required to obtain the symmetry-protected TATP are broken, while the conditions needed to define q are kept, we obtain **d**. Notice that the nodal ring (black) formed between the upper two bands are penetrated by two nodal lines formed between the lower two bands. This should be compared with **e**, in which we do not obtain a linked nodal ring structure as in **d**, although we similarly perturb the Hamiltonian in **c**.

where $n = 1, 3, 4, 6, 8$ and $\lambda_n$ are the Gell–Mann matrices. For cubic groups, we find $f_1(\mathbf{k}) = ak_xk_y$, $f_3(\mathbf{k}) = b(k_x^2 - k_y^2)$, $f_4(\mathbf{k}) = ak_xk_z$, $f_6(\mathbf{k}) = ak_yk_z$, $f_8(\mathbf{k}) = \frac{b}{\sqrt{3}}(k_x^2 + k_y^2) - \frac{2b}{\sqrt{3}}k_z^2$. Here, $a$ and $b$ are constants that can be related to the three elastic constants of a cubic crystal, $C_{11}$, $C_{12}$, and $C_{44}$, by the following relations: $a = C_{12} + C_{44}$ and $b = \frac{C_{11}}{2} - \frac{C_{44}}{2}$. When $a \neq 0$, the topological properties of the Hamiltonian are determined by only one parameter, $b/a$, so that we can draw a phase diagram as shown in Fig. 3a. We find that $b/a > 0$ corresponds to the band structure shown in Fig. 1a, so that the $L$ mode is gapped from the $T$ modes for $k > 0$, and the topological charge is q = (1, 2). When $b/a < 0$, q is not defined because it is no longer possible to properly partition the energy bands for $k > 0$ (see Fig. 3b, c).

The criterion $b/a > 0$ allows us to easily search for materials with TATP. Applying this criterion to some monatomic lattices, we find that the acoustic phonons in Au, Ag, and Cu are topological with q = (1, 2). Since monatomic lattices have only three phonon modes, the nonzero topological charge q diagnosed from the elastic continuum approximation does not reduce to e even if we consider the full Hamiltonian.

It turns out that the above condition $b/a > 0$ that the phonons carry q amounts to the condition that the longitudinal velocity exceeds the transverse velocity along the high-symmetry lines (see Supplementary note 6). For isotropic systems, the transverse velocity cannot exceed the longitudinal velocity because of the Born stability condition for isotropic systems that $v_T^2/v_L^2 < 3/4$. However, the Born stability criteria of cubic crystals[37,38] do not forbid $v_T > v_L$ along the high-symmetry lines so that it is possible to observe acoustic phonons that do not carry q. The necessary and sufficient conditions for stability of cubic crystals are[39] $C_{44} > 0$, $C_{11} - C_{12} > 0$, $C_{11} + 2C_{12} > 0$, which allows $v_T > v_L$. Indeed, such situations are known to occur[40] in certain Tm-Se

and Sm-Y-S intermediate valence compounds[41,42] and certain Mn-Ni-C alloys[43,44].

Although TATP can appear for any crystal symmetry for acoustic phonons, the symmetry-protected TATPs of phonons, or of electrons, require stricter symmetry constraints. Of the 32 point group symmetries, only the $O_h$ and the $T_h$ groups contain the inversion symmetry and support three-dimensional representations. In the case of the $O_h$ group (see Supplementary Note 4 for $T_h$ group), four representations ($T_{1u}$, $T_{2u}$, $T_{1g}$, $T_{2g}$) allow a triple point, and the k.p Hamiltonian near the triple point takes the form in Eq. (2) after appropriate transformations. Therefore, the phase diagram in Fig. 3 applies here as well.

Finally, let us note that although we have used the approximate continuum Hamiltonian in the above discussion, it is possible to obtain a three-band Hamiltonian for the three bands constituting the TATP in multiband systems up to any desired accuracy by using the Löwdin partitioning (for details, see TATP in multiband systems in Methods). This allows us to define the TATP without relying on the approximate continuum Hamiltonian even in multiband systems.

**Nodal structure.** The charge q = ($n_{sk}$, e) strongly constrains the nodal structure. As before, we consider a sphere on which q is nontrivial. First, because the skyrmion number of the $L$ mode cannot change under a continuous deformation of the Hamiltonian without closing the gap, the $L$ mode must cross the $T$ modes inside the sphere, which occurs at the ATP. Second, the Euler number constrains the number of nodal lines formed between the $T$ modes that pass through the ATP. This is because nodal lines emanating from the TATP can be considered as Dirac points on the 2D sphere surrounding the TATP, and nonzero e constrains the total vorticity $N_t$ (signed count of the number of Dirac points)

to be $N_t = -2\mathfrak{e}$[18]. Thus, when $\mathfrak{e} = 2$ for the $T$ modes, $N_t = -4$, so that there must be at least four nodal lines emanating from the TATP (see Vorticity and nodal lines in Methods).

At this point, it is interesting to note that the presence of symmetry-protected TATPs requires more constraints than it is needed to define $\mathfrak{q}$. This is because the definition of $\mathfrak{q}$ only requires that the Hamiltonian be a $3 \times 3$ real-symmetric matrix with a spectral gap between the $L$ and the $T$ modes, while the symmetry-protected TATP requires further constraints such as the $O_h$ symmetry. Thus, it is natural to ask how the topological charge $\mathfrak{q} = (1, 2)$ constrains the nodal structure of symmetry-protected TATPs when we perturb the Hamiltonian so that the relevant symmetry is relaxed, while the conditions required to define $\mathfrak{q}$ are maintained. Let us note that we allow the possibility for the perturbation to break the $\mathcal{P}$ and the $\mathcal{T}$ symmetries, but the perturbation must preserve the combined $\mathcal{PT}$ symmetry to keep the Hamiltonian real. In Fig. 3d, we show the nodal structure that results from adding such perturbations to the Hamiltonian used in Fig. 3b. As explained further in Linked nodal structure protected by $\mathfrak{q}$ in Methods, we find a nodal ring formed between the $L$ and the $T$ modes (black ring) that is threaded by two nodal lines formed between the $T$ modes (red lines). For comparison, we similarly perturb the Hamiltonian for the case where $\mathfrak{q}$ is ill defined, used in Fig. 3c. The resulting nodal structure is shown in Fig. 3e. Although the nodal structure is complicated, we see that there is no linking structure similar to that observed in Fig. 3d.

Let us note that although we discussed how $\mathfrak{q}$ constrains the nodal structure by focusing on three-band systems here, our conclusion can be generalized to multiband systems as in (see Continuum theory in Results) by using the Löwdin partitioning[45] (see TATP in multiband systems in Methods for details).

**Avoiding the doubling theorem**. It is well known that Weyl points must appear in pairs because of the Nielsen–Ninomiya theorem[46], which is simply the result of the periodicity of the BZ and the topological charge (Chern number) that protects the Weyl points. In the case of TATP, we can expect that the Euler number $\mathfrak{e}$, which characterizes the two $T$ modes of TATP, can play a similar role as the Chern number. That is, we can expect that the $\mathfrak{e}$ computed on a 2D slice of the BZ will change by $\pm 2$ across a TATP, so that the TATP should occur in pairs to keep the periodicity of the BZ. However, the doubling theorem can be avoided in various ways. Here, we give two representative examples to avoid the doubling of TATP. Since $\mathfrak{e}$ is the quantity that is needed in the above argument for the doubling, let us recall that $\mathfrak{e}$ on a 2D BZ can be defined only if the two bands that characterize the TATP are separated from all the other bands by an energy gap on the 2D BZ, and only if the Zak phases for these two bands vanish for all non-contractible loops. In the two examples we give below, the doubling theorem is avoided because one of the two conditions needed to define $\mathfrak{e}$ on a 2D slice of the BZ is not satisfied.

As the first example, let us consider the phonon spectrum of CsCl in Fig. 2b. First, let us note that there is a gap between the lowest three phonon modes (acoustic phonons) and the rest of the phonon modes (optical phonons). Focusing on the acoustic phonons, we see that there are two triple points at $\Gamma$ and $R$. However, the triple point at $\Gamma$ is topological while that at $R$ is not. This is allowed because there is no energy gap between the $L$ and the $T$ modes along $RM$. Because this nodal line stretches across each of the $k_x$, $k_y$, and $k_z$ directions, it is not possible to choose a 2D plane in the BZ such that the $L$ and the $T$ modes are gapped consistently. Therefore, it is not possible to define $\mathfrak{e}$ for the lowest two bands on any 2D plane.

Interestingly, the doubling theorem can be avoided even when there is a gap between the $L$ and the $T$ modes in the whole BZ except at a TATP. This is because $\mathfrak{e}$ is defined only for an orientable vector bundle. The obstruction to orientability is given by the $\pi$ quantized Zak phase computed along a non-contractible loop in the BZ[18], which is just the 1D topological invariant in $\mathcal{PT}$-symmetric systems. In other words, if the Zak phase is nontrivial along a non-contractible loop in the 2D slice of the BZ, the Euler number cannot be defined. Therefore, a single TATP can appear in the presence of $\pi$ quantized Zak phase for the $T$ modes because the periodicity argument cannot be carried out.

We demonstrate this with the second example. Here, we consider the electronic spectrum of the 3D generalization of the Lieb lattice[47,48], whose lattice structure is shown in Fig. 4a (see Supplementary Note 7 for the details). From the resulting band structure and the Wilson loop spectrum in Fig. 4b, c, we see that there is a single TATP at $R$, although the $L$ and the $T$ modes are fully gapped except at $R$. We numerically confirmed that there is $\pi$ quantized Zak phase for the $T$ modes along each of the $k_x$, $k_y$, and $k_z$ direction, which allows a single TATP.

**Discussion**

For phonons in cubic crystals, either $\mathfrak{q} = (1, 2)$ or $\mathfrak{q}$ is undefined. However, when the symmetry of the crystal is sufficiently low, it is also possible to obtain $\mathfrak{q} = (0, 0)$. The acoustic phonon of tellurium with space group $P3_121$ is one such example, as we illustrate in Fig. 5. Here, let us note that tellurium lacks the inversion symmetry, so that strictly speaking, $\mathfrak{q}$ is not defined. However, in the elastic continuum approximation, the inversion symmetry is restored as explained in Theory of elastic continuum in Methods, and this does not nullify our discussion. We show the gap closing points in the acoustic phonon spectrum in Fig. 5b. Notice that the gap closing points occur only between the $T$ modes, so that we can define $\mathfrak{q}$. From the winding structure of the Wilson loop spectrum shown in Fig. 5c, we see that $\mathfrak{e} = 0$, which is consistent with the trivial skyrmion texture of $\mathfrak{e}_{\mathbf{k},L}$, see Fig. 5d. Since the TATP in the tellurium has $\mathfrak{q} = (0, 0)$, there are no nontrivial links in the nodal structure when a perturbation that eliminates the TATP but preserves the $\mathcal{PT}$ symmetry is added to the dynamical matrix (Fig. 5e).

Because topological charge is often associated with surface states, it is natural to ask whether there are relevant surface states. Since surface acoustic wave is well-known surface states related to acoustic phonons, one may suspect that it is related to $\mathfrak{q}$. For an isotropic medium, the stability of the material imposes the condition $v_T^2/v_L^2 < 3/4$, while the condition for the appearance of surface acoustic waves is $v_T^2/v_L^2 < 1$. Therefore, isotropic elastic materials satisfying the stability condition always have surface acoustic waves[33]. However, because isotropic phonon is topological even for $v_T^2/v_L^2 > 1$, topology does not seem to be directly related to the surface-localized states. To further confirm this, we study the finite size 3D Lieb lattice model. As we discuss in detail in Supplementary Note 8, we find that even when the parameters are chosen so that the k.p theory for the TATP at $R$ becomes the same as the elastic continuum theory for the isotropic phonon, there are no surface-localized states. Because the same continuum theory does not lead to the same boundary states, we conclude that surface acoustic waves result from the boundary condition specific to elastic systems.

Although nontrivial $\mathfrak{q}$ is not directly related to surface states, it can induce anomalous transport phenomena, such as the phonon angular momentum Hall effect. In Ref. [49], it was observed that the winding structure of isotropic phonon has a characteristic phonon angular momentum Hall response. Here, phonon angular momentum refers to the angular momentum generated by an atom in a lattice as it orbits about its equilibrium position, and the

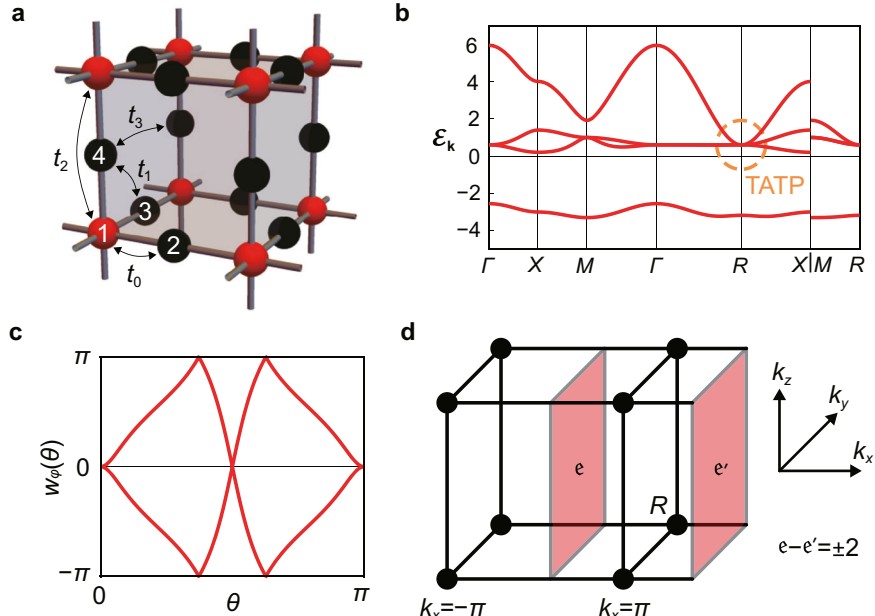

**Fig. 4 Avoiding the doubling theorem in 3D electronic Lieb model. a** 3D Lieb lattice structure with four sites in a unit cell. **b** The electronic band structure with a single TATP at $R$. Here $\Gamma = (0, 0, 0)$, $X = (\pi, 0, 0)$, $M = (\pi, \pi, 0)$, and $R = (\pi, \pi, \pi)$. We refer to the highest energy mode forming the TATP as the $L$ mode, and the lower two energy modes forming the TATP as the $T$ modes. **c** Wilson loop spectrum for the second and third lowest bands, corresponding to $T$ mode of the TATP, computed over a sphere with radius $0.1\pi$ centered at $R$. The winding structure shows that $|e| = 2$. **d** With a single TATP at $R$ (black dots), it is not possible to define $e$ in the $(k_y, k_z)$ plane since it conflicts with the periodicity of the Brillouin zone. This contradiction is resolved by noticing the $\pi$ Zak phases along the $k_x$, $k_y$, $k_z$ directions, so that $e$ is ill defined.

phonon angular momentum Hall effect refers to the flow of phonon angular momentum in the direction perpendicular to the temperature gradient. As a consequence of the phonon angular momentum Hall effect, there is an edge accumulation of phonon angular momentum, which can have significant contributions from both the bulk and surface-localized phonons (see Supplementary Note 9 for details). In ionic crystals, such surface-localized phonon angular momentum can induce surface magnetization, which is a measurable quantity. Here we find that the winding structure of the phonon polarization, which drives the phonon angular momentum Hall effect in Ref. [49], arises from the topological charge $q = (1, 2)$. Because the phonon angular momentum Hall effect and the orbital Hall effects are analogous, the TATPs at high-symmetry points consisting of $p$ or $d$ electron orbitals are also a significant source of the orbital Hall effect. The relation between the orbital Hall effect and the orbital texture characterized by $q = (1, 2)$ was discussed in detail in Refs. [50,51], although the relation between $q$ and the orbital texture was overlooked. Further investigating the physical consequences of having different topological characterizations of TATPs will be an interesting topic for future study.

## Methods

**Note on terminology**. Here, we clarify some of the terminology used in this work. By ATP, we refer to the triple degeneracy of acoustic phonons that arises due to the NG theorem. When an ATP carries the nontrivial topological charge $q$, we refer to it as a TATP. In this case, the TATP occurs because of the NG theorem, but it is also possible to obtain a triple degeneracy at high-symmetry points due to the point group symmetry. When such a triple degeneracy carries the nontrivial topological charge $q$, it is also referred to as a TATP, as discussed in the main text (see Continuum theory in Results).

The term continuum Hamiltonian is used to refer to the approximate low-energy Hamiltonian for a set of bands of interest. To avoid confusion, when we discuss the continuum Hamiltonian for the acoustic phonons, we use the term elastic continuum Hamiltonian, see Theory of elastic continuum in Methods. Note also that when we consider the isotropic elastic continuum, we always specify the isotropicity. When we discuss the continuum Hamiltonian for the symmetry-protected triple degeneracy, we use the term k.p Hamiltonian. Note that it is

important to distinguish the k.p Hamiltonian from the elastic continuum Hamiltonian because the symmetry constraints are not the same, see the discussion on $T_h$ group in Supplementary Note 4.

**Homotopy description of** $q$. Here, we give a homotopy description of the topological charge $q$. As in the main text, we consider the $3 \times 3$ real-symmetric $H_k$ at a fixed $k > 0$, with the TATP situated at $k = 0$. Because there is a spectral gap between the $L$ mode and the $T$ modes for $k > 0$ (note that there are two $T$ modes, $T_1$ and $T_2$), the Hamiltonian can be written as

$$H_{\mathbf{k}} = E_{\mathbf{k}}^T \begin{pmatrix} 1 & 0 & 0 \\ 0 & -1 & 0 \\ 0 & 0 & -1 \end{pmatrix} E_{\mathbf{k}}, \quad E_{\mathbf{k}} = \begin{pmatrix} \epsilon_{\mathbf{k},L} \\ \epsilon_{\mathbf{k},T_1} \\ \epsilon_{\mathbf{k},T_2} \end{pmatrix}, \quad (3)$$

after a spectral flattening in which the eigenvalues of the $L$ and the $T$ modes are sent to 1 and $-1$, respectively. Since $E_{\mathbf{k}} \in O(3)$ and the Hamiltonian is invariant under $E_{\mathbf{k}} \to F_{\mathbf{k}} E_{\mathbf{k}}$ with $F_{\mathbf{k}} \in O(1) \times O(2)$, the topological charge of the triple point at $k = 0$ can be characterized by the second homotopy group[34] of the classifying space $B = O(3)/[O(1) \times O(2)]$, which is $\pi_2(B) = 2\mathbb{Z}$. In Supplementary Note 1, we use the exact sequence of homotopy groups for fibration to show explicitly that this topological charge is 2 for isotropic phonons. As further discussed in Supplementary Note 1, this charge can be shown to be equivalent to the topological charge $q = (n_{sk}, e)$ defined in the main text, where $e = 2n_{sk}$, see also Refs. [52–54].

**Computation of Euler number using Wilson loop**. The absolute value of the Euler number $|e|$ can be computed by the using the Wilson loop spectrum on a sphere surrounding the TATP. To compute the Wilson loop spectrum, let us define the $2 \times 2$ overlap matrix $[F_j]_{mn} = \epsilon_{\mathbf{k}_j,m} \cdot \epsilon_{\mathbf{k}_j,n}$, where $m, n \in \{T_1, T_2\}$, and $\mathbf{k}_j = k(\sin\theta\cos\phi_j, \sin\theta\sin\phi_j, \cos\theta)$ where $\phi = 2\pi j/N$ for some integer $N$. The Wilson loop operator at $\theta$ is defined as $W_\phi(\theta) = \lim_{N \to \infty} F_{N-1} F_N ... F_1 F_0$. Defining $w_\phi(\theta)$ to be the imaginary part the eigenvalues of $\ln W_\phi(\theta)$, we can compute $|e|$ by counting the number of times $w_\phi(\theta)$ crosses $\pi$.

In the case of isotropic phonons, the transverse modes are tangent to the sphere on which $q$ is defined, so that $e = 2$, and the Wilson loop spectrum shows the double winding structure. For isotropic phonons, there is an alternative explanation to this double winding structure in the Wilson loop spectrum. Because $H_{\mathbf{k}}$ is invariant under the $SO(2)$ rotation symmetry about the axis $\hat{\mathbf{k}}$, we can split the eigenstates according to the eigenvalues of the helicity operator $\hat{\mathbf{k}} \cdot \mathbf{L}$, where $\mathbf{L} = (L_x, L_y, L_z)$ is the spin-1 matrix representation of angular momentum. Because the helicity is $\pm 1$ for the transverse modes, we can define the Chern numbers for the transverse modes in the helicity sectors, which are $\mp 2$ (see Supplementary Note 2). Since the Wilson loop spectrum for a band with Chern number $n$ shows $n$

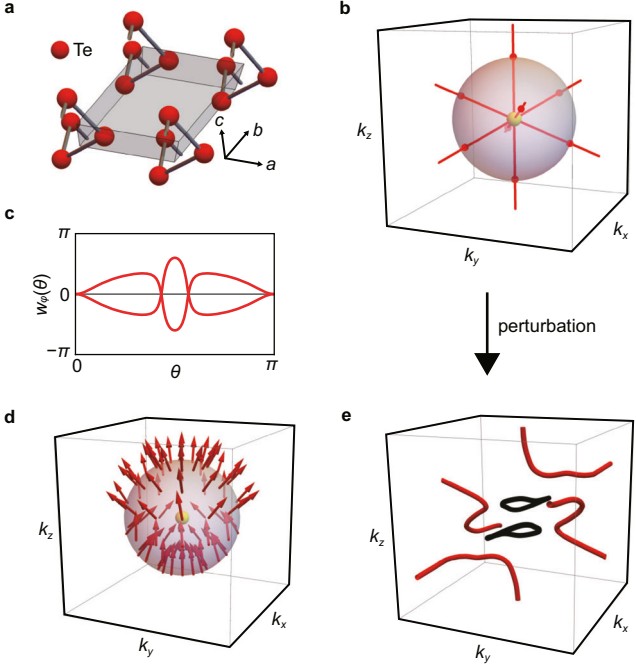

**Fig. 5 Acoustic phonon of tellurium. a** The lattice structure of tellurium with space group P3₁21. **b** By using the values of the stiffness tensor in Materials Project[68], we find that away from the triple point, the red nodal lines occur only between the two lowest energy bands so that q can be defined. **c** The Wilson loop spectrum shows trivial winding structure, and therefore, $\mathfrak{e} = 0$. **d** The wavefunction texture of the highest energy band shows trivial skyrmion texture, which is consistent with the constraint that $2\mathfrak{n}_{sk} = \mathfrak{e}$. **e**, We perturb the dynamical matrix such that the degeneracy of TATP in **b** is lifted (note, however, that these perturbations break the continuous translation symmetry, since the triply degenerate Goldstone mode disappears.). The black nodal lines formed by the two highest energy bands are created. As opposed to the case with nontrivial q in Fig. 3d, the nodal structure in **e** does not have link between the red and black nodal lines because q is zero.

chiral windings, we see that there should be two branches with opposite winding in the Wilson loop spectrum, corresponding to the two helicity sectors with opposite Chern numbers.

**Vorticity and nodal lines.** We can define the vorticity of a Dirac point when the Hamiltonian has the $\mathcal{PT}$ symmetry. The effective Hamiltonian around a Dirac point can be written as a $2 \times 2$ real-symmetric matrix, $H_D = r(\mathbf{k})\cos\theta(\mathbf{k})\sigma_x + r(\mathbf{k})\sin\theta(\mathbf{k})\sigma_z$. The vorticity is defined as the winding number of $(\cos\theta(\mathbf{k}), \sin\theta(\mathbf{k}))$ around the Dirac point. Although the vorticity can easily be defined locally around the Dirac point, its global definition is nontrivial. A careful analysis[18] shows that a two-band insulator with Euler number $\mathfrak{e}$ has even number of Dirac points such that the total sum of their vorticity $N_t$ satisfies the relation $-\frac{N_t}{2} = \mathfrak{e}$.

This can be directly applied to the TATP: because the Euler number for the transverse acoustic phonons is 2, there must be a minimum of four Dirac points on a sphere surrounding the ATP, such that the total sum of their vorticity is $-2\mathfrak{e}$. As we change the radius of this sphere, the trajectories of the Dirac points form nodal lines, so that there must be a minimum of four nodal lines emanating from the TATP. Because two nodal lines emanating from the TATP can smoothly be connected, there must be a minimum of two nodal lines passing through the TATP (i.e., a nodal line emanating from the TATP is one half of a full nodal line passing through the TATP).

**Linked nodal structure protected by** q. Let us explain why the symmetry-protected TATP evolves into a nodal ring threaded by two nodal lines when the symmetry that protects the triple degeneracy is relaxed. First, $\mathfrak{n}_{sk}$ requires the gap between the $L$ mode and the $T$ modes to close inside the sphere on which q is defined. However, because the triple point is no longer protected, and the generic nodal structure in a real-symmetric Hamiltonian in 3D is the nodal line, the gap closing points between the $L$ mode and the $T$ modes evolve into a nodal ring, see Fig. 3d (see Supplementary Note 5 for the details of the Hamiltonian). Second, $\mathfrak{e}$

requires at least four Dirac points to form between the $T$ modes on the sphere on which q is defined. Equivalently, at least two nodal lines formed between the $T$ modes must pass through this sphere. As can be seen in Fig. 3d, these two nodal lines formed between the $T$ modes (red lines) penetrate the nodal ring formed between the $L$ and the $T$ modes (black ring). Such a structure is required because otherwise it is possible for the nodal ring to be gapped out after deforming to a point, which is not compatible with the charge $\mathfrak{n}_{sk} = 1$ of the $L$ mode. We provide a simple geometric proof of this property in Supplementary Note 5, and we note that a similar observation was also made in Ref. [55] using quaternion charges.

**Details of ab initio calculations.** For the computation of the band structure and Wilson loop spectrum of the phonons in CsCl, we employed the Vienna ab initio simulation package (VASP)[56] with the projector augmented-wave method[57]. The generalized gradient approximation (PBE-GGA) is employed for exchange-correlation potential[58]. We used the default VASP potentials (Cs_sv and Cl), and a 500 eV cutoff. To get the force constant, a $6 \times 6 \times 6$ supercell and a $6 \times 6 \times 6$ Monkhorst-pack k-point mesh were used. The phonon eigenvalues and the eigenstates were calculated using the PHONOPY package[59]. The dynamical matrix and the force constants were obtained from the frozen phonon method, based on the Hellmann–Feynman theorem.

Let us note that this calculation does not take into account the non-analytic correction terms to the dynamical matrix[60]. Since the optical phonons in ionic insulators such as CsCl can be strongly renormalized by the non-analytic correction terms[61–63], the stability of the symmetry-protected ATPs requires a more thorough analysis.

**Theory of elastic continuum.** Acoustic waves in crystals can be well described by the elastic continuum theory. The elastic continuum theory[33,36] is an excellent approximation in the long wavelength limit, where the variation in the displacement $\mathbf{u}$ occurs over a length scale of $10^{-6}$ cm or larger for typical crystals. When a crystal is time-reversal symmetric, the elastic continuum is described by the Lagrangian density, $\mathcal{L}[\mathbf{u}, \dot{\mathbf{u}}] = T[\dot{\mathbf{u}}] - U[\mathbf{u}]$, where $T[\dot{\mathbf{u}}]$ and $U[\mathbf{u}]$ denote the kinetic energy density and elastic energy density, respectively. The kinetic energy is given by $T[\dot{\mathbf{u}}] = \frac{1}{2}\rho\dot{u}^2$ where $\rho$ is the mass density. The elastic energy density is proportional to the square of strain tensor $u_{ij} = \frac{1}{2}(\partial_i u_j + \partial_j u_i)$ where $\partial_i = \partial/\partial x_i$: $U[\mathbf{u}] = \frac{1}{2}\lambda_{ijlk}u_{ij}u_{kl}$, where $\lambda_{ijl}$ is the elastic modulus tensor. Then, the equation of motion is given by $\rho\ddot{u}_i = \partial_j(\lambda_{ijkl}u_{kl})$. Fourier transformation of this equation yields $D(\mathbf{k})_{ij}u_j(\mathbf{k}) = \omega_\mathbf{k}^2 u_i(\mathbf{k})$, which is nothing but the eigenvalue equation for the phonon spectrum $\omega_\mathbf{k}$. (Further details on the elastic continuum can be found in Supplementary Note 3.) Here, the dynamical matrix $D(\mathbf{k})_{ij}$ is defined as $D(\mathbf{k})_{ij} = \rho^{-1}\lambda_{iljm}k_l k_m$ and it is referred to as the "Hamiltonian" $H_\mathbf{k}$ in the main text.

From this dynamical matrix, we see that for ATPs formed by acoustic phonons in a crystal with time-reversal symmetry, the topological charge q can be defined for both centrosymmetric and non-centrosymmetric elastic crystals as long as the elastic continuum limit is considered. Recall that a necessary condition to define q is that the Hamiltonian must be a real-symmetric $3 \times 3$ matrix, where the reality condition can be satisfied if there is $\mathcal{PT}$ symmetry. Since we are assuming that $\mathcal{T}$ is a symmetry of the crystal, the above statement holds if $\mathcal{P}$ is a symmetry of the elastic continuum Hamiltonian, and this is precisely the case even for non-centrosymmetric crystals.

Now, let us explain why this statement is true, since it is essential for characterizing the acoustic phonons with q. First, notice that under $\mathcal{P}$, we have $\mathbf{u}(\mathbf{k}) \rightarrow -\mathbf{u}(-\mathbf{k})$. Then, the constraint on the $D(\mathbf{k})$ from $\mathcal{P}$ is $D(\mathbf{k}) = D(-\mathbf{k})$, which is obviously true because $D(\mathbf{k})$ is quadratic in $\mathbf{k}$, even in non-centrosymmetric crystals. Consequently, $\mathcal{PT}$ is a symmetry of the elastic continuum theory, and this allows us to define q. Notice that this argument is true whether or not the crystal is centrosymmetric or non-centrosymmetric. On the other hand, $\mathcal{P}$-breaking terms are allowed when we consider terms that are higher-order in $\mathbf{k}$. Nevertheless, such higher-order terms are negligible for the description of acoustic phonons with long wavelengths. To conclude, the topological charge q can be defined as local property of acoustic phonons in time-reversal symmetric crystals.

Next, let us comment if $\mathcal{T}$ is broken in the phonon Hamiltonian, it may not be restored in the elastic continuum limit, which is in contrast to the behavior of $\mathcal{P}$. Typically, the time-reversal breaking terms in the phonon Hamiltonian is modeled by terms such as the Raman spin-phonon coupling[64], whose leading contribution is constant in $\mathbf{k}$, and the Mead–Truhlar term in the Born–Oppenheimer approximation[65,66], whose leading contribution is quadratic in $\mathbf{k}$. Because these terms do not vanish in the elastic continuum limit, $\mathcal{T}$ is not restored in the elastic continuum limit.

To summarize, $\mathcal{P}$ (and therefore $\mathcal{PT}$) is a symmetry of the elastic continuum Hamiltonian in $\mathcal{T}$-symmetric crystals, but $\mathcal{T}$ symmetry is generally not restored as a symmetry in the elastic continuum Hamiltonian in $\mathcal{T}$-broken crystals.

**TATP in multiband systems.** The Löwdin partitioning[45] is a method used to divide a system into two mutually non-interacting subsystems. Based on the Löwdin partitioning, let us first show how q can be defined for TATPs in $\mathcal{PT}$ symmetric multiband systems by extracting a $3 \times 3$ Hamiltonian for the TATP by decoupling it from the rest of the energy bands. We will then show how our results

on the linked nodal structure (see Nodal structure in Results), which was a consequence of nonzero q in a three-band system, can also be extended to multiband systems.

Let us begin by considering an $N \times N$ Hamiltonian $H(\mathbf{k}) = H(0) + \delta H(\mathbf{k})$. We can assume that the TATP occurs at $\mathbf{k} = 0$ and that $H(0)$ is block diagonal,

$$H(0) = \begin{pmatrix} H_A(0) & 0_{(3,N-3)} \\ 0_{(N-3,3)} & H_B(0) \end{pmatrix}, \tag{4}$$

where $0_{(n,m)}$ denotes the $n \times m$ zero matrix. The subsystem $A$ contains the TATP and is described by the $3 \times 3$ matrix $H_A(0)$, and the subsystem $B$, which is separated from the subsystem $A$ by an energy gap, is described by the $(N - 3) \times (N - 3)$ matrix $H_B(0)$. By treating $\delta H(\mathbf{k})$ perturbatively, the Löwdin partitioning block diagonalizes $H(\mathbf{k})$ into an effective $3 \times 3$ Hamiltonian for the subsystem $A$ and an effective $(N - 3) \times (N - 3)$ Hamiltonian for the subsystem $B$ up to any order in $\mathbf{k}$ through a unitary transformation $e^{-S(\mathbf{k})}H(\mathbf{k})e^{S(\mathbf{k})}$, see Ref. [67] for the explicit form of $S(\mathbf{k})$. This partitioning can be carried out for $\mathbf{k}$ such that the gap $\Delta E$ between $A$ and $B$ is large, i.e., $\delta H(\mathbf{k}) \ll \Delta E$, which can always be satisfied for small enough $\mathbf{k}$. We thus see that even in multiband systems, we can systematically extract a $3 \times 3$ Hamiltonian that describes the three bands that comprise the TATP.

Now, let us consider the effect of a perturbation that breaks the triple degeneracy of TATP while maintaining the $\mathcal{PT}$ symmetry. Such a perturbation was used when we studied the nodal structure constrained by q in the main text. With the goal of carrying out the Löwdin partitioning, we can express the perturbation as $\lambda V(\mathbf{k}) = \lambda V(0) + \lambda \delta V(\mathbf{k})$ where $\lambda$ is a parameter that controls the strength of the perturbation. The full Hamiltonian is then given by $H_\lambda(\mathbf{k}) = H(\mathbf{k}) + \lambda V(\mathbf{k})$, and we can apply the Löwdin partitioning in a similar way to the unperturbed case. In this way, we can systematically obtain a $3 \times 3$ Hamiltonian $H_{A,\lambda}(\mathbf{k})$ for the subsystem A as a function of $\lambda$ as long as $\delta H(\mathbf{k}) + \lambda \delta V(\mathbf{k}) \ll \Delta E$. We thus see that the evolution of the nodal structure as we turn on the perturbation can be described exactly as if we were working with a strictly three-band system.

## Data availability

The data that support the findings of this study are available from the corresponding author upon reasonable request.

## Code availability

The code that supports the findings of this study is available from the corresponding author upon reasonable request.

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

## Acknowledgements

S. P. thanks Sunje Kim for useful discussion. S. P., Y. H. and B.-J. Y. were supported by the Institute for Basic Science in Korea (Grant No. IBS-R009-D1), Samsung Science and Technology Foundation under Project Number SSTF-BA2002-06, and the National Research Foundation of Korea (NRF) Grant funded by the Korea government (MSIT) (No. 2021R1A2C4002773 and No. NRF-2021R1A5A1032996). H. C. C was supported by the Institute for Basic Science in Korea (Grant No. IBS-R009-D1).

## Author contributions

S.P. initially conceived the project. S.P. and Y.H. equally contributed to the theoretical analysis and wrote the manuscript with B.-J.Y. H.C.C. did all of the ab initio calculations. B.-J.Y. supervised the project. All authors discussed and commented on the manuscript.

## Competing interests

The authors declare no competing interests.
