## [Peer Review File · Nature Communications]

Topological acoustic triple pointREVIEWER COMMENTS

Reviewer #1 (Remarks to the Author):

In this work, the authors discuss a topological invariant associated with certain triply degenerate nodal points (TDNP) in band structures of 3D translationally-symmetric matter. On the one hand, TDNPs can be stabilized at high-symmetry points in the momentum space of effectively spinless particles in cubic or tetrahedral lattices; on the other hand, Nambu-Goldstone (NG) theorem imposes TDNPs in the acoustic (as well as optical) phonon spectra also at high-symmetry points of the momentum space. Due to a mathematical analogy, the authors consider both options under the single umbrella notion of "acoustic triple point" (ATP).

The authors characterize these three-fold degeneracies using a topological invariant "q", which is a combination of the skyrmion number and of the Euler class on a sphere surrounding the degeneracy. If the invariant "q" is non-trivial, the degeneracy is dubbed "topological acoustic triple point" (TATP). Implications of this invariant for the nodal-line degeneracies of the band structure are considered, and relevance to certain anomalous transport phenomena is briefly foreshadowed in the conclusions.

The possibility to characterize NG modes by a topological invariant is certainly an interesting observation. In particular, while the possibility to characterize the longitudinal mode by a skyrmion number is intuitively clear and expected, the authors put these ideas on a solid mathematical footing, reveal valuable analogies to electron band structures, and also demonstrate on the case of elemental tellurium that the mentioned skyrmion number can (to me quite unexpectedly) even be trivial.

For these reasons, I find that the work potentially contains enough original results to warrant acceptance in journal Nature Communications.

However, I also must state that I find the presently submitted text at many places inadequately structured, as if written in a hurry. In multiple places, the assumptions and the logical reasoning leading to the results are obscured to a level that I failed to follow. Thus, the authors should take care to considerably improve the overall organization of the manuscript before I can give my final assessment.

I also have several clarifying questions that the authors should consider in their resubmission. I hope these concrete questions will help the authors identify the problematic points of their presentation:

- 1.) The manuscript has a relatively long supplementary information file (SIF). I think the readers would benefit if (a) a table of contents were given in the beginning of SIF, if (b) the main-text references to SIF referred to its specific section (S1 to S9), and if (c) more material were moved from SIF to Methods (subject to the limitations of the journal) to increase its visibility.
- 2.) The role of inversion (P) symmetry is unclear at several places. I understand that PT symmetry is needed to define the Euler class, yet the PT symmetry is somehow restored in the continuum limit $k \rightarrow 0$ of even non-centrosymmetric crystals. Could the authors elaborate on this aspect? (Does this statement remain true in magnetic crystals?)
- 3.) The authors discuss acoustic spectrum of isotropic media at several places. However, no crystalline media with finite unit cell constant (not even the cubic ones) are isotropic, and I find the corresponding discussions brings more confusions than clarifications -- especially since this notion is often considered alongside the continuum limit. I urge the authors to treat the discussion of the isotropic limit (and the inequalities on the elastic tensor it entails) with more care. In fact, I am wondering if the isotropic limit is needed at all for any of the derivations presented by the authors.
- 4.) The topological invariant is specified as $q=(n_{sk},e)$ where " n_{sk} " is the skyrmion number defined

only in the 3-band models/limit, and "e" is the Euler number which in 3-band models obeys $e=2n_{sk}$. It therefore seems that the definition " $q=(n_{sk},e)$ " is redundant; in principle, "e" contains all the topological information (irrespective of the number of bands). Why do the authors not choose this seemingly simpler definition of the topological invariant?

5.) Concerning the last paragraph of Sec.II: First, it refers to a discussion of optical TATPs in the Methods section (which I wasn't able to locate). Second, concerning the R-point, why not split the bands into the top two and the bottom one (instead of the other way round)? I assume that a similar flipping has to be performed for the "symmetry-protected TATPs" arising in electron band structures with cubic/tetrahedral symmetry.

6.) In Sec.III the authors write that "the acoustic phonons of monoatomic lattices [have] $q=(1,2)$ ", but I fear I overlooked the proof of this statement.

7.) The evolution of ATP into a nodal link visible in Fig.3(d) seems similar to a similar transformation of 3DNP into a nodal link as discussed by Ref.[26]. Could the authors comment if there is a relation between these two works? Also, it would be interesting to see the analogous data for the case where "q" is trivial, such as for tellurium in Fig.5 (perhaps through an extra panel).

8.) This question is more my curiosity and does not need to be reflected in the manuscript (or even answered in the authors' reply): When describing the dynamics of a lattice, one can theoretically add a potential term " $V = \sum_{\{a,i\}} k * u_{\{a,i\}} * u_{\{a,i\}} / 2$ " where " $u_{\{a,i\}}$ " is the displacement of atom at site "a" from the equilibrium position in direction "i". (This means that we first identify the equilibrium position of each atom, and then bind the atom to this position with a string of stiffness "k") Such a term breaks the translation symmetry of the Hamiltonian (since the atoms acquire a preferred lowest-energy positions in position space), and therefore gaps out the NG modes. I am wondering if the Euler class and its relation to the linking of nodal lines still apply in such a setting.

9.) Some clarification of the "phonon angular momentum Hall effect" should be added to the main text. Note that the abstract advertises that "the [TATPs] can induce anomalous thermal transport in phononic systems and orbital Hall effect in electronic systems"; however, I find these statements inadequately reflected in the manuscript's main text.

10.) I would suggest two potentially relevant references for the authors' considerations. First, when comparing the Hamiltonian description + topological characterization of phonon vs. electron systems, a reference to <https://doi.org/10.1073/pnas.1605462113> might be appropriate. Second, when considering the possible topological origin of the surface acoustic waves, a reference to the work <https://arxiv.org/abs/2004.09517> on Rayleigh edge states in certain chiral crystals could be relevant.

Reviewer #2 (Remarks to the Author):

This work shows that the triple degeneracy of acoustic phonons can be characterized by a topological charge q . The topological charge q can equivalently be characterized by the skyrmion number of the longitudinal mode, or by the Euler number of the transverse modes. The authors did stop there, and so the authors proceeded to study how TATP strongly constrains the nodal structure around the TATP when there is a perturbation. Investigating topological phonon band structures, particularly the triple points in the phonon bands, is of considerable interest, but at this point I am not sure if this work has completed an in-depth study that warrants publication in Nature Communications. This reviewer also finds the writing style, which is intended to be rather concise, contains confusing statements here and there and hence weaken my overall impression on this work.

One page 1, left column and start of right column, authors say "The topological charge q is strictly defined only when the total number of energy bands is fixed to there". This is followed by "The TATP

protected by NG theorem exists ubiquitously in elastic material.”

Question 1: I am confused by these two self-contradicting statements. The first sentence said clearly that we strictly need 3 bands only, yet the second sentence says that there is no restriction on the number of phonon bands.

On page 2 (left column), the authors say “This condition is strictly satisfied by the phonons in a monatomic lattice, which have only three phonon bands. Even when there is more than one atom per unit cell, and therefore more than three phonon energy bands, this condition is satisfied near the ATP, which can be described by the elastic continuum Hamiltonian”

Question 2: The confusion is similar. If the second half of the sentence is true, then it seems that there is no need to emphasize the condition of only having three bands in many places of this manuscript. This style of the writing is creating an unpleasant/insecure experience. Taking the second point from the authors -- for systems with more than 3 bands, one must be reminded again that only local behaviour of the TATP can be discussed. Can the authors confirm that later, when they talk about the perturbation to the system and then the nodal line structure, did they really restrict themselves in the near TAPT regime?

Question 3: In a few places, the authors stressed the role of PT symmetry. But for real symmetric matrices under consideration, the time-reversal symmetry is always there. So what is the point of considering the joint PT symmetry, which is only about P-symmetry given that T-symmetry is already present? In my impression, the authors never spells out what the actual role is as played by PT symmetry (and yet they emphasize this throughout the writing). This is a concern.

Question 4: The authors investigated how TATP charge q constrains the nodal structure of symmetry-protected TATPs when we perturb the Hamiltonian so that the relevant symmetry is relaxed, while the conditions required to define q are maintained. While some interesting consequences are observed due to TATP, there is no connection between the topological features of the resultant nodal structure (such as some topological invariants depicting the linkage of the nodal lines) and the TATP charge q . Based on the authors' reasoning, this connection seems to be possible but was not done.

Question 5: In section 5 “AVOIDING THE DOUBLING THEOREM”, the authors discussed some examples of how this is achieved. However, the authors did not really establish theoretically why this well-known theorem is not relevant to TATPs ---exactly what part of the assumption of this theorem is violated? Given many qualitative analyses in the manuscript, the lack of a solid reason to account for the avoiding of the doubling theorem is a bit disappointing to this reviewer.

Reviewer #3 (Remarks to the Author):

In this paper, Park et al have established the topological nature of a special "node" in the band of phonons, which is the zero-momentum-zero-energy nexus of the triplet Goldstone modes. It is shown that a pair of topological invariants, the Euler number e and the Skyrmion number q , describe the above topology. Similar discussion applies to triplet node at higher energy protected by cubic symmetries. Possible consequences for this type of topology are discussed briefly.

For someone who is familiar with Ref.[35] from the same group, and an earlier paper from another group which was not cited, the above result adds limited information to the existing literature. The work at hand is more or less an application of the existing theory to the acoustic phonons. For that reason, I think the work is not seminal enough as a theoretical breakthrough. On the other hand, I appreciate the idea that a rather abstract Z_2 topological invariant (which is the Euler number mod 2) has found its realization in one of the most common phenomenon of acoustic waves that appear in

almost every solid. Such identification, ideally, should come with new physical effect to be experimentally observed, yet the discussion of possible effects originating from the new topology is too brief to be convincing or helpful for experimentalists. For that reason, I think the work has not enough potential to drive new experiments. Based on the above observations, I conclude that the physics shown in the current form is not sufficiently interesting for publication in Nature Communications.

Sec.I and Sec.II:

In earlier works, it has been shown that $\pi_2(O(M+N)/O(M)*O(N))=Z^2$ for $M, N > 2$, and also that for $N=2$ (or $M=2$), this invariant becomes a Z -invariant (later named as the Euler number). Here the authors are dealing with $M=1, N=2$, so a straightforward calculation would show that $e=2$, or $z^2=1$.

Sec.III:

This section is good. It gives clear criterion when the topological invariant becomes nontrivial in terms of the elastic constants, and relates this criterion to the stability criterion. One cannot derive such information trivially from existing results.

It is interesting to see, however, that some phonon bands carry $q=0$. Is it possible to have $q > 1$? If so I would find it very interesting.

Sec.IV:

In Ref.[35], authors from the same group give very comprehensive discussion of what happens as the nodal point breaks into a nodal line, and the linking structure that is basically the same as Fig.3d. Rather, in Ref.[35], the discussion is even more general than here, because they were dealing with arbitrary M and N . Simply applying the general results to $M=1, N=2$, one immediately finds the linking structure.

A related question is what happens if $q > 1$ ($e > 2$)? Should one expect a more complicated linking structure than Fig.3d?

Sec.V:

This section is interesting. It involves the difference between the definition of topological invariants on S^n and that on T^n . I suggest to expand this section, specially where the authors claim that such an extension to T^n is possible if and only if the non-contractible loop has trivial Berry phase.

Sec.IV:

I suggest that the authors significantly expand the discussion on experimental consequences. For me, several points should be clarified. (i) Since the nodal point is related to PT -symmetry, which is not preserved on any open surface, can I say that there is no topologically protected surface states related to this type of node? (ii) Since phonons do not couple to gauge fields, how should one observe the phonon angular momentum Hall effect? (iii) Can we have a concrete calculation of the orbital Hall effect in an electronic material hosting TP at Fermi energy?

The bottom line is that since the topological part of the theory is not new enough, the authors may consider shifting the emphasis to potentially new observable effects.

Reviewer #1 (Remarks to the Author):

In this work, the authors discuss a topological invariant associated with certain triply degenerate nodal points (TDNP) in band structures of 3D translationally-symmetric matter. On the one hand, TDNPs can be stabilized at high-symmetry points in the momentum space of effectively spinless particles in cubic or tetrahedral lattices; on the other hand, Nambu-Goldstone (NG) theorem imposes TDNPs in the acoustic (as well as optical) phonon spectra also at high-symmetry points of the momentum space. Due to a mathematical analogy, the authors consider both options under the single umbrella notion of "acoustic triple point" (ATP).

The authors characterize these three-fold degeneracies using a topological invariant "q", which is a combination of the skyrmion number and of the Euler class on a sphere surrounding the degeneracy. If the invariant "q" is non-trivial, the degeneracy is dubbed "topological acoustic triple point" (TATP). Implications of this invariant for the nodal-line degeneracies of the band structure are considered, and relevance to certain anomalous transport phenomena is briefly foreshadowed in the conclusions.

The possibility to characterize NG modes by a topological invariant is certainly an interesting observation. In particular, while the possibility to characterize the longitudinal mode by a skyrmion number is intuitively clear and expected, the authors put these ideas on a solid mathematical footing, reveal valuable analogies to electron band structures, and also demonstrate on the case of elemental tellurium that the mentioned skyrmion number can (to me quite unexpectedly) even be trivial.

For these reasons, I find that the work potentially contains enough original results to warrant acceptance in journal Nature Communications.

Authors' response: We indeed thank the reviewer for positively evaluating our work.

However, I also must state that I find the presently submitted text at many places inadequately structured, as if written in a hurry. In multiple places, the assumptions and the logical reasoning leading to the results are obscured to a level that I failed to follow. Thus, the authors should take care to considerably improve the overall organization of the manuscript before I can give my final assessment.

I also have several clarifying questions that the authors should consider in their resubmission. I hope these concrete questions will help the authors identify the problematic points of their presentation:

Authors' response: We thank the reviewer for carefully reading our manuscript and giving us valuable comments. The reviewer's comments helped us strengthen our logic and improve the presentation style in our manuscript. In the following, we answer all of the questions and comments raised by the reviewer.

1.) The manuscript has a relatively long supplementary information file (SIF). I think the readers would benefit if (a) a table of contents were given in the beginning of SIF, if (b) the main-text references to SIF referred to its specific section (S1 to S9), and if (c) more material were moved from SIF to Methods (subject to the limitations of the journal) to increase its visibility.

Authors' response: Following the reviewer's suggestion, we revised the manuscript and the supplementary information file (SIF) in the following way.

(a) We added a table of contents at the beginning of SIF, where we also briefly introduce which materials are covered in the SIF.

(b) We specified the relevant section in the SIF for each of the references to SIF in the main text.

(c) Because the contents in the SIF are mostly proofs and details of the calculations, we could not find a suitable way to move the contents of the SIF to the Methods section. Instead, we have added some new materials in the Methods. These new sections are titled "Note on terminology", "Theory of elastic continuum", and "TATP in multiband systems".

2.) The role of inversion (P) symmetry is unclear at several places. I understand that PT symmetry is needed to define the Euler class, yet the PT symmetry is somehow restored in the continuum limit $k \rightarrow 0$ of even non-centrosymmetric crystals. Could the authors elaborate on this aspect? (Does this statement remain true in magnetic crystals?)

Authors' response: We thank the reviewer for this question, which helped us clarify some of the subtle points regarding the symmetry in our manuscript. As the reviewer mentions, the space-time-inversion (PT)

symmetry is necessary to define the topological charge q composed of the Euler class e and the skyrmion number π_{sk} . In general, in a system lacking the inversion symmetry or the time reversal symmetry, the inversion and time-reversal symmetries are not restored in the continuum limit. On the other hand, the continuum limit of acoustic phonons is special because it can be well described by the theory of elasticity. Then, even in non-centrosymmetric crystals, the inversion symmetry (P) is restored in the continuum limit. On the other hand, in time-reversal symmetry broken crystals, the time-reversal symmetry (T) may not be restored in the continuum limit. To clarify this, let us first explain why P is restored in the continuum limit in crystals which breaks the P symmetry but do not break the T symmetry.

Elastic crystals are described well by the continuum limit when deformations in crystals have long wavelength (typically larger than 10^{-6} cm). In the continuum limit, dynamics of low-energy excitations in elastic crystals is determined by the elastic energy and kinetic energy. The elastic energy is quadratic in the strain tensor (a spatial derivative of displacement \mathbf{u}). On the other hand, the kinetic energy is quadratic in $\dot{\mathbf{u}}$ (a time derivative of \mathbf{u}). Then, the equations of motion and its Fourier transforms can be obtained according to the Hamilton's principle. In this way, we can obtain the relation between the dynamical matrix $D(\mathbf{k})$ and dispersion relation $\omega(\mathbf{k})$ of low-energy excitation. Explicitly, this relation is $D(\mathbf{k})_{ij} u_j(\mathbf{k}) = \omega(\mathbf{k})^2 u_i(\mathbf{k})$ where $u_i(\mathbf{k})$ is the i -th component of displacement \mathbf{u} after Fourier transformation and i denotes the spatial direction. Given this equation, the symmetries in continuum limit are determined by those of the dynamical matrix $D(\mathbf{k})$. Since the elastic energy is quadratic in the strain tensor, the dynamical matrix $D(\mathbf{k})$ is quadratic in momentum \mathbf{k} . That is, all the elements of dynamical matrix are quadratic in momentum. Now, let us note that P acts as follows: $\mathbf{u}(\mathbf{k}) \xrightarrow{P} -\mathbf{u}(-\mathbf{k})$. Therefore, the constraint of P is $D(\mathbf{k}) = D(-\mathbf{k})$, which is trivially satisfied because $D(\mathbf{k})$ is quadratic in \mathbf{k} . Consequently, P is restored in continuum limit even for non-centrosymmetric crystals *as long as* the elastic energy is considered up to quadratic order in the strain tensor. However, if we depart from elastic continuum limit by considering higher-order terms in \mathbf{k} , P may not be restored. However, these higher-order terms are negligible for the acoustic modes in the elastic continuum limit.

In contrast to the inversion symmetry, the terms used to describe the time-reversal breaking effects in phonons do not disappear in the elastic continuum limit (this answers the reviewer's question on the magnetic crystal). Notable examples of time-reversal breaking interactions are the constant Raman spin-phonon coupling (discussed for example in [L. Zhang et al., PRL 105, 225901 (2010)]) and the Mead-Truhlar term in the Born-Oppenheimer approximation (discussed for example in PRB 86, 104305 (2012) and PRL 123, 255901 (2019)). Both are corrections to the phonon Hamiltonian of the form $(\mathbf{p}(\mathbf{k}) - \mathbf{a}(\mathbf{u}(\mathbf{k})))^2$, where the former is a model of the form $\mathbf{a} \propto \mathbf{u}(\mathbf{k})$, and the latter is a model of the form $\mathbf{a} \propto k^2 \mathbf{u}(\mathbf{k})$. Since these coupling terms arise because of the external magnetic field, they break the time-reversal symmetry, which can also be checked by noting that under the time reversal, $\mathbf{p}(\mathbf{k}) \rightarrow -\mathbf{p}(-\mathbf{k})$, $\mathbf{u}(\mathbf{k}) \rightarrow \mathbf{u}(-\mathbf{k})$. Insofar as these models for time-reversal breaking effects are correct, we see that time-reversal breaking terms remain important even in the long-wavelength limit. Therefore, the Hamiltonian in the continuum limit has neither T symmetry nor the PT symmetry. For such time-reversal breaking effects, we cannot define the topological charge q even in the continuum limit.

In summary, for acoustic phonons in time-reversal-symmetric crystals, the inversion symmetry is present in the continuum limit even when the crystal is not centrosymmetric. This enable us to define the topological charge q that requires PT symmetry. However, in crystals with broken time-reversal symmetry, the time-reversal symmetry may not be restored in the continuum limit, even for phonons.

In the revised manuscript, we clarified these points by adding the section titled "theory of elastic continuum" in Methods.

3.) The authors discuss acoustic spectrum of isotropic media at several places. However, no crystalline media with finite unit cell constant (not even the cubic ones) are isotropic, and I find the corresponding discussions brings more confusions than clarifications -- especially since this notion is often considered alongside the continuum limit. I urge the authors to treat the discussion of the isotropic limit (and the inequalities on the elastic tensor it entails) with more care. In fact, I am wondering if the isotropic limit is needed at all for any of the derivations presented by the authors.

Authors' response: As the reviewer pointed out, the isotropic limit is not essential for deriving any of our main results. For example, the topological charge q can be defined without mentioning the isotropic continuum.

Nevertheless, we choose to discuss the isotropic limit frequently because it is both simple and well-known. Furthermore, it allows us to explain in simple language how the topological charge q is defined, and also to derive some exact results on the eigenstates and surface acoustic waves. These exact results are expected to help readers understand our main results. For instance, by using the isotropic limit, we can easily find the exact

condition for the appearance of the surface acoustic waves ($\frac{v_T^2}{v_L^2} < 1$), which we invoke in the discussion section to conclude that surface acoustic waves are not directly related to the topology of the ATP.

On the other hand, we agree that the terminology in our manuscript can be confusing. To clarify this, we have decided to change all references to the continuum Hamiltonian of acoustic phonons to “elastic continuum Hamiltonian.” We also checked that when we consider isotropic phonons, the isotropicity is explicitly specified. We have also changed the references to the continuum Hamiltonian near symmetry enforced triple degeneracy to “k.p Hamiltonian”. To further eliminate sources of confusion, we have added a section titled “Note on terminology” in Methods.

4.) The topological invariant is specified as "q=(n_{sk},e)" where "n_{sk}" is the skyrmion number defined only in the 3-band models/limit, and "e" is the Euler number which in 3-band models obeys e=2n_{sk}. It therefore seems that the definition "q=(n_{sk},e)" is redundant; in principle, "e" contains all the topological information (irrespective of the number of bands). Why do the authors not choose this seemingly simpler definition of the topological invariant?

Authors' response: As the reviewer mentioned, the Euler number is defined for multi-band systems, in contrast to the skyrmion charge n_{sk} , which is defined only for three-band systems (To be more precise, n_{sk} becomes trivial when there are more than three bands in the following sense: let us note that n_{sk} is just the second homotopy group $\pi_2(S^2)$, characterized by the maps S^2 (sphere in momentum space) $\rightarrow S^2$ (wavefunction space). We can generalize the definition of n_{sk} to systems with more than three bands by considering maps $S^2 \rightarrow S^n$ with $n > 2$. However, these maps are contractible to a point: $\pi_2(S^n) = 0$ for $n > 2$). Also, in three-band systems, the skyrmion charge n_{sk} is given by a half of the Euler number e , as we derived a relation $e = 2n_{sk}$ in the manuscript. Because of this relation, our definition $q = (n_{sk}, e)$ can seem redundant. However, these two topological charges (n_{sk} and e) are quite different in nature, and hence they play different roles of constraining the nodal structure.

First, the skyrmion charge n_{sk} is classified as a delicate topological invariant in our system because it is trivialized even when we add topologically trivial bands with arbitrary energies. On the other hand, the Euler number e corresponds to fragile topological invariant. Following the recent classification scheme of topological insulators, we think that it is better to distinguish between n_{sk} and e , even if they are related in a three-band system. Thus, our convention of expressing the topological charge $q = (n_{sk}, e)$ as a tuple of two different classes of topological invariants clarifies that $q = \pi_2(O(3)/O(1) \times O(2))$ is delicate topological charge and that q is reduced to the Euler number e when additional bands are added.

Second, n_{sk} helps us understand the nodal structure of TATP in Section IV. Especially, when the triple degeneracy of TATP are lifted by a PT-preserving perturbation, a nodal ring formed by the L mode and one T mode must have a nontrivial linking structure with another nodal ring formed by the two T modes. While the Euler number constrain the total winding number of vortices which corresponds to intersection points of nodal lines formed by the T modes and a sphere encircling the original TATP, the skyrmion charge is needed to explain the linking structure.

Finally, the choice $q = (n_{sk}, e)$ clarifies that we are considering the topological charge of a three-band system. If we instead use only e to specify the topological charge, it can be confusing especially when we discuss systems having more than three bands. This is because the charge $q = (1,2)$ of a TATP in a multiband system is defined only in the continuum limit (or by Löwdin partitioning), and this reduces to $e = 2$ if we stray away from the continuum limit, see for example the discussion in Sec. V “Avoiding the doubling theorem”. It would become more difficult to distinguish between these two viewpoints if we did not introduce a separate notation for the topological charge in a three-band system.

5.) Concerning the last paragraph of Sec.II: First, it refers to a discussion of optical TATPs in the Methods section (which I wasn't able to locate). Second, concerning the R-point, why not split the bands into the top two and the bottom one (instead of the other way round)? I assume that a similar flipping has to be performed for the "symmetry-protected TATPs" arising in electron band structures with cubic/tetrahedral symmetry.

Authors' response: We thank the reviewer for giving these constructive comments. Let us first note that reference was to Methods F (Details of ab initio calculations) in the revised manuscript, where we comment that optical phonons in insulating ionic crystals can be strongly renormalized, which is usually modeled by non-analytic corrections to the dynamical matrix. Our theory assumes the Hamiltonian to be at least continuous, so that it cannot be applied to such a non-analytic effective Hamiltonian. We did not thoroughly investigate this problem because it is a completely different problem in nature to the one we are trying to solve, and the focus in the manuscript was on the acoustic modes. In the revised manuscript, we clarify which section was being referenced.

Now, we answer the reviewer's second question on how to split the bands. In general, there are two ways to split the one 'longitudinal' and two "transverse" modes of TATP: we can split the three bands into 1) one highest band and two lowest bands or 2) one lowest band and two highest bands. The topological charge q can be defined only when one of the two gap conditions is maintained over the entire sphere wrapping the ATP.

However, the triple points at high-symmetry point do not always satisfy these two gap conditions. As can be seen in Fig. 2b, the bands near R point satisfy neither of the aforementioned gap conditions. In high-symmetry line R-M, two highest bands are degenerate. On the other hand, two lowest bands are nearly degenerate along two high-symmetry lines R-X and R- Γ . Consequently, we cannot consistently partition the energy bands around the triple degeneracy into 2-to-1 fashion, and we cannot assign it the topological charge q .

This comment on R point was in Sec. V, but our comment on R point in Sec. II "...because lower two bands cannot be fully separated from the highest energy band" can be a bit misleading. We have therefore revised this comment as follows: "...because it is not possible to consistently separate one of the energy bands from the other two on a sphere surrounding the triple degeneracy"

6.) In Sec.III the authors write that "the acoustic phonons of monoatomic lattices [have] $q=(1,2)$ ", but I fear I overlooked the proof of this statement.

Authors' response: We apologize for causing the confusion. The original statement was "The criterion $b/a > 0$ allows us to easily search for materials with TATP. In particular, the acoustic phonons of monatomic lattices such as Au, Ag, and Cu are topological with $q = (1,2)$." By this, we meant that the criterion $b/a > 0$ allows us to easily find materials with TATP, and that by using this criterion, we can easily find that Au, Ag, and Cu are *examples* of monatomic lattices hosting TATP.

In the revised manuscript, we have revised this expression as follows: "Applying this criterion to some monatomic lattices, we find that the acoustic phonons in Au, Ag, and Cu are topological with $q = (1,2)$. Since monatomic lattices have only three phonon modes, the nonzero topological charge q diagnosed from the elastic continuum approximation does not reduce to e even if we consider the full Hamiltonian."

7.) The evolution of ATP into a nodal link visible in Fig.3(d) seems similar to a similar transformation of 3DNP into a nodal link as discussed by Ref.[26]. Could the authors comment if there is a relation between these two works? Also, it would be interesting to see the analogous data for the case where " q " is trivial, such as for tellurium in Fig.5 (perhaps through an extra panel).

Authors' response: We thank the reviewer for this insightful question. In Ref. [26], the authors consider the triple points formed by a band inversion of singly degenerate band and doubly degenerate bands. The double degeneracy of the latter is protected by rotation and the PT symmetry. Since the triple points originate from the band inversion, they must be formed in pairs. On the other hand, triple points in acoustic systems (i.e. ATP) originate from the spontaneous breaking of translation symmetry, and it does not need particular symmetries such as rotation symmetry. Also, when TATP in cubic symmetric system is located at high-symmetry point such as R point, the triple degeneracy of TATP corresponds to three-dimensional irreducible representation of cubic symmetry group. Hence, the protecting symmetry is clearly different from that of Ref. [26].

On the other hand, if we compare Fig. 2d in Ref. [26] (panel that shows the linking structure arising from breaking the rotational symmetry that protects the type-A triple points) with Fig. 3d in our manuscript, the linking structure is similar. Then, one may ask: if we collide two type-A triple points in Ref. [26], will the two triple points reduce to the one in our manuscript? We found that a recent preprint by the same authors (arXiv:2104.11254) mentions this problem, which we summarize here: The answer to the above question cannot be determined just from the linking structure, because of the possibility that the two type-A triple points will pair-annihilate, and open the gap between the uppermost band and the lower two bands in Fig. 2 of Ref. [26]. On the other hand, it can also happen that the two type-A triple points cannot be pair-annihilated. In this case, after fine-tuning the Hamiltonian parameters to bring the two triple points together, the resulting triple point will have the same topological charge as the one we study in our manuscript. Conversely, if we take the symmetry-protected TATP in our manuscript and reduce the cubic symmetry group to C_4 or C_{4v} (while maintaining the PT symmetry), we should obtain a pair of type-A triple points that shows a topologically protected linking structure upon breaking the rotational symmetry that protects the triple degeneracy.

Finally, let discuss the nodal structure of the triple points with trivial q as in the case of tellurium. In our revised manuscript, the right panel of Fig.5 (subfigures b and e) illustrates the evolution of the nodal structure when $q = (0,0)$, which is to be compared with the evolution of nodal structures in the case when $q = (1,2)$ and when q is not well-defined (Fig. 3). As can be seen, there are no linking structures when $q = (0,0)$. For convenience, we reproduce Fig. 5e below. Let us note, however, that such a perturbation necessarily breaks the continuous symmetry, since the triply degenerate Goldstone modes no longer appears.

8.) This question is more my curiosity and does not need to be reflected in the manuscript (or even answered in the authors' reply): When describing the dynamics of a lattice, one can theoretically add a potential term " $V = \sum_{\{a,i\}} k \cdot u_{\{a,i\}}^2 / 2$ " where " $u_{\{a,i\}}$ " is the displacement of atom at site "a" from the equilibrium position in direction "i". (This means that we first identify the equilibrium position of each atom, and then bind the atom to this position with a string of stiffness "k") Such a term breaks the translation symmetry of the Hamiltonian (since the atoms acquire a preferred lowest-energy positions in position space), and therefore gaps out the NG modes. I am wondering if the Euler class and its relation to the linking of nodal lines still apply in such a setting.

Authors' response: In the situation the reviewer considered, the triple degeneracy at zero momentum does not exist in general. However, our results on the topological characterization and linking structure of nodal lines can still be applied to this situation: As long as the additional potential term $V = \sum_{a,i} \frac{K}{2} u_{ai}^2$ with spring constant K preserves PT symmetry, the topological charge q can be defined and constrains the linking structure of the nodal structure arising from breaking the triple degeneracy. In fact, this situation is analogous to triple points at high-symmetry points. In Section IV, we discuss the nodal structure when the degeneracy of triple point at high-symmetry points is lifted by a PT preserving perturbation. The potential term V considered by the reviewer plays the same role as such PT preserving perturbation.

9.) Some clarification of the "phonon angular momentum Hall effect" should be added to the main text. Note that the abstract advertises that "the [TATPs] can induce anomalous thermal transport in phononic systems and orbital Hall effect in electronic systems"; however, I find these statements inadequately reflected in the manuscript's main text.

Authors' response: We thank the reviewer for this comment. The reason that we did not discuss the phonon angular momentum Hall effect and the orbital Hall effect in detail in the main text was due to the reason that they were investigated in detail in Refs. [51,52,53], although their connection to the topological charge was overlooked in these references. Since many readers may not be familiar with the relation between the wavefunction texture and the anomalous transport, we have slightly expanded the discussion related to these phenomena.

10.) I would suggest two potentially relevant references for the authors' considerations. First, when comparing the Hamiltonian description + topological characterization of phonon vs. electron systems, a reference to <https://doi.org/10.1073/pnas.1605462113> might be appropriate. Second, when considering the possible topological origin of the surface acoustic waves, a reference to the work <https://arxiv.org/abs/2004.09517> on Rayleigh edge states in certain chiral crystals could be relevant.

Authors' response: We thank the reviewer for bringing our attention to interesting references [R. Süsstrunk and S. D. Huber, PNAS.1605462113] and [C. Benzoni et al., arXiv:2004.09517].

The first reference [R. Süsstrunk and S. D. Huber, PNAS.1605462113] discusses topological classification of mechanical systems. In mechanical systems they considered, continuous translation symmetry is not broken spontaneously as in phonons. Therefore, the band structure is gapped in general. The authors classified these systems in a similar way that topological insulators are classified. In particular, their classification scheme uses the well-known Altland-Zirnbauer (AZ) symmetry classes to characterize the stable topology of mechanical systems that is robust under the addition of topologically trivial bands.

However, we would like to point out that the method employed in the PNAS paper to relate the electron and phonon problem is different from ours because the authors consider a more general problem: $\ddot{x}(t) = \sum_{j=1}^N [-D_{ij} x(t) + \Gamma_{ij} \dot{x}(t)]$. This requires them to treat the phonon Hamiltonian as a $2N$ by $2N$ matrix, where N

is the number of energy modes. In contrast, we consider the case when $\Gamma_{ij} = 0$, which is usually the case in phonons (if we do not consider phonon damping, Raman-like terms, and Mead-Truhlar term in the Born-Oppenheimer approximation). This allows us to restrict our attention to just the dynamical matrix D . Also, the authors of the paper discuss the edge modes related to the stable topological charge. On the other hand, the topological charge we discuss is the delicate charge, so that the theory discussed in the PNAS paper cannot be applied to our problem.

In the second reference [C. Benzoni et al., arXiv:2004.09517], the authors investigate the Rayleigh wave in two-dimensional elastic solid with broken time-reversal symmetry. They conclude that because of the time-reversal breaking term, which they call the “Berry term”, the Rayleigh modes can appear asymmetrically, and even merge into the bulk modes, and that this can be controlled by the Poisson ratio. Like us, the authors of this paper conclude that the Rayleigh modes they discuss are not of topological origin (although the authors make this claim, we could not locate their argument regarding this point). We would like to note, that the symmetry class discussed in this paper is different from the one in our manuscript, so that the discussion there cannot be directly applied to the situation in our manuscript.

Reviewer #2 (Remarks to the Author):

This work shows that the triple degeneracy of acoustic phonons can be characterized by a topological charge q . The topological charge q can equivalently be characterized by the skyrmion number of the longitudinal mode, or by the Euler number of the transverse modes. The authors did stop there, and so the authors proceeded to study how TATP strongly constrains the nodal structure around the TATP when there is a perturbation. Investigating topological phonon band structures, particularly the triple points in the phonon bands, is of considerable interest, but at this point I am not sure if this work has completed an in-depth study that warrants publication in Nature Communications. This reviewer also finds the writing style, which is intended to be rather concise, contains confusing statements here and there and hence weaken my overall impression on this work.

Authors' response: We thank the reviewer for the valuable questions and comments, which helped us improve parts of the manuscript which the reviewer has found confusing, and which could also have caused confusion for other readers. As the reviewer said, there is indeed a great interest in the topology of phonon band structure, and particularly about triple points in phonons. In this context, we think that our result that acoustic phonons can be characterized by a delicate topological charge, which is often deemed artificial by the community that studies topological insulators and semimetals, can be of wide interest. Unfortunately, the reviewer felt that our work has not completed an in-depth study. However, we think this impression was made not due to the lack of progress made in our work, but rather due to some of the statements that the reviewer found confusing, which also seems to be the reason that the reviewer thought that some of the statements were analyzed only qualitatively. In our revised manuscript, we have carefully considered the questions and comments raised by the reviewer to improve the overall presentation of our results. We hope the reviewer will find that our revised manuscript is suitable for publication in Nature Communications.

One page 1, left column and start of right column, authors say “The topological charge q is strictly defined only when the total number of energy bands is fixed to there”. This is followed by “The TATP protected by NG theorem exists ubiquitously in elastic material.”

Question 1: I am confused by these two self-contradicting statements. The first sentence said clearly that we strictly need 3 bands only, yet the second sentence says that there is no restriction on the number of phonon bands.

Authors' response: We agree that the statement can seem contradictory. We think this confusion was caused because we have not specified the context in which the statement holds in the introduction section (please also refer to our response to the reviewer's Question 2). To clarify the meaning of this statement, we have reworded the second statement as follows: “Although only phonons in monatomic lattices have precisely three-bands, the theory of elasticity naturally yields an effective three-band description of the ATP, which are the three NG modes. In this sense, we find that TATP protected by the NG theorem is ubiquitous in elastic materials.”

On page 2 (left column), the authors say “This condition is strictly satisfied by the phonons in a monatomic lattice, which have only three phonon bands. Even when there is more than one atom per unit cell, and therefore more than three phonon energy bands, this condition is satisfied near the ATP, which can be described by the elastic continuum Hamiltonian.”

Question 2: The confusion is similar. If the second half of the sentence is true, then it seems that there is no need to emphasize the condition of only having three bands in many places of this manuscript. This style of the writing is creating an unpleasant/insecure experience. Taking the second point from the authors -- for systems with more than 3 bands, one must be reminded again that only local behaviour of the TATP can be discussed. Can the authors confirm that later, when they talk about the perturbation to the system and then the nodal line structure, did they really restrict themselves in the near TAPT regime?

Authors' response: We thank the reviewer for asking this question. Especially, the question about the linking structure in *multiband* systems, which we have not considered in the previous version of the manuscript, helped us generalize the statement in our manuscript.

We agree that the statement the reviewer is pointing to can seem contradictory if the reader did not catch the point we wanted to make about the elastic continuum Hamiltonian. Although the reviewer may already be familiar with the theory of elasticity, let us first recall the basics of the theory of elasticity to clarify the point we wanted to make in our manuscript. Applied to crystals, the theory of elasticity assumes that the displacement $\mathbf{u}_{n,\alpha}$ of atom at unit cell index n and sublattice α (so that the equilibrium position is $\mathbf{x}_{n,\alpha}$) can be considered as a

continuous vector field $\mathbf{u}(\mathbf{x})$. This allows us to define the strain tensor $u_{ij} = \frac{\partial u_i}{\partial x_j} + \frac{\partial u_j}{\partial x_i}$ and the elastic free energy $F = \frac{1}{2} \sum_{ijkl} \lambda_{ijkl} u_{ij} u_{kl}$. As is well known, this is an excellent approximation for acoustic waves, and it naturally yields a 3-by-3 dynamical matrix. In other words, even in crystals with many optical phonons, the long-wave dynamics (the three acoustic waves) can be described by an effective 3-by-3 dynamical matrix, and this approximation is commonly made when studying the properties of acoustic waves.

The point here is that the dynamical matrix studied in elastic continuum theory is a 3-by-3 matrix, and it is an excellent approximation for acoustic waves. Furthermore, the dynamical matrix is a *real-valued* 3-by-3 matrix regardless of the number of optical phonons, and our theory can be applied whenever the elastic continuum theory is employed (Note that by stating that the dynamical matrix is ‘real-valued’, we are implicitly assuming that the time reversal symmetry is not broken in the phonon spectrum. Under this assumption, the dynamical matrix in the continuum limit is real valued. Please refer to our response to the reviewer 1’s question 2 about this point on time reversal symmetry in elastic continuum limit.) In other words, although our theory is applicable only for strictly three-band systems (which can occur in monatomic lattices), effective Hamiltonians that contain only three bands arise naturally through the theory of elasticity, and our theory is also applicable to these Hamiltonians as long as these approximate Hamiltonians are valid. We note however, that even in the case when the three-band description is an approximation to the full phonon Hamiltonian, it is still meaningful in the sense that the Euler number in the topological charge $q = (\pi_{sk} \epsilon)$ is well defined even in systems with more than three bands. Thus, if the elastic continuum theory (which is exact in the limit as $\mathbf{k} \rightarrow 0$) predicts $q = (1,2)$, it implies that $q = (1,2) \rightarrow \epsilon = 2$ if we do not employ the continuum approximation, but consider the full phonon Hamiltonian.

At this point, the reviewer may also ask about the TATPs that are not described by the elastic continuum theory, but instead appear at high-symmetry points due to the symmetry constraints, and their status when there are more than three bands in total. A similar discussion applies in these cases. Locally, these TATP can be described well through the k.p Hamiltonian near the TATP, which becomes exact in the limit $\mathbf{k} \rightarrow 0$, and the charge q is well-defined in this limit. On the other hand, if we depart from k.p theory, q is not well defined because the skyrmion number is not well defined in a multiband system. However, the Euler number is still well-defined, and in this sense, q is still meaningful as discussed above.

Let us now answer the question about the nodal line structure resulting from perturbations that break the triple degeneracy. In the previous version of the manuscript, we have proved that linking structure will emerge only for a 3-by-3 matrix in the Supplementary Information (Supplementary Note 5 “Linking structure protected by q ”). Because this proof holds only for a strictly three-band system, we have discussed the linking structure only for strictly three-band systems in our manuscript. On the other hand, we have found that near the TATP, it is not difficult to extend our proof for the three-band system to systems with more than three bands, as long as a gap between the three bands in question and the other energy bands is maintained, as we show below through an application of the Löwdin perturbation theory, which is also known as Löwdin partitioning [P. Löwdin, J. Chem. Phys. 19, 1396 (1951)].

First, let us consider the case when we do not apply perturbation that breaks the triple degeneracy. We can write the N-by-N Hamiltonian as $H(\mathbf{k}) = H(0) + \delta H(\mathbf{k})$, where we can assume without loss of generality that the TATP occurs at $\mathbf{k} = 0$ and that $H(0)$ is block diagonal:

$$H(0) = \begin{pmatrix} A(0) & 0 \\ 0 & B(0) \end{pmatrix}.$$

Here, A block is the 3-by-3 matrix that contains the triple degeneracy, and B is the (N-3)-by-(N-3) matrix that contains all the other energy bands. Then, for \mathbf{k} near 0, it is possible to obtain an effective 3-by-3 Hamiltonian for the A block and (N-3)-by-(N-3) Hamiltonian for the B blocks through a unitary transformation $e^{-S(\mathbf{k})} H(\mathbf{k}) e^{S(\mathbf{k})}$, where $S(\mathbf{k})$ can be expanded perturbatively in $(\delta H / \Delta E)$, where ΔE is the (smallest) energy gap between A and B blocks, see [Appendix B] in [R. Winkler, Spin-orbit coupling effects in two-dimensional electron and hole systems, (Springer, 2003)] for explicit form of $S(\mathbf{k})$. Therefore, as long as the perturbation theory converges (for \mathbf{k} for which gap between A and B blocks is large), there exists a basis that can be obtained through the unitary transformation $e^{S(\mathbf{k})}$ such that the Hamiltonian can be expressed as a block diagonal matrix with two diagonal blocks corresponding to A and B blocks. Then, the Hamiltonian for the three bands hosting the TATP is given by the first diagonal block, which is a 3-by-3 matrix. Note that this 3-by-3 Hamiltonian obtained from the Löwdin partitioning describes the band structure about TATP exactly, in contrast to the continuum approximation obtained by small \mathbf{k} expansion.

Now, let us consider the effect of perturbations that breaks the triple degeneracy. This perturbation can similarly be written as $\lambda V(\mathbf{k}) = \lambda V(0) + \lambda \delta V(\mathbf{k})$, where λ controls the strength of the perturbation. Applying the above perturbation theory to $H_\lambda(\mathbf{k}) = H(\mathbf{k}) + \lambda V(\mathbf{k})$, we can obtain an effective 3-by-3 Hamiltonian for the three bands corresponding to the TATP, as long as $(\delta H + \lambda \delta V)/\Delta E$ is small.

In this way, we see that although we have proved the linking structure in the presence of perturbations only for strictly three-band system, it can be applied to cases when there are more than three bands as long as the gap between the TATP and the other bands is large.

In the revised manuscript, we have added this argument in the section titled ‘‘TATP in multiband systems’’ in Methods.

Question 3: In a few places, the authors stressed the role of PT symmetry. But for real symmetric matrices under consideration, the time-reversal symmetry is always there. So what is the point of considering the joint PT symmetry, which is only about P-symmetry given than T-symmetry is already present? In my impression, the authors never spells out what the actual role is as played by PT symmetry (and yet they emphasize this throughout the writing). This is a concern.

Authors’ response: We thank the reviewer for pointing out that we have not spelled out why the PT symmetry is needed to impose the reality condition in the main text (we found that the reason was explained only in the Supplementary Information). Before explaining the role of the PT symmetry, let us first note that time reversal symmetry alone does not constrain the Hamiltonian to be real and symmetric, which the reviewer seems to be claiming. In fact, the time reversal symmetry does not have to be present in a real symmetric Hamiltonian. A simple one-band Hamiltonian (or energy spectrum) that shows this in 1D is $H(k) = \sin(k) + \cos(k)$. This real Hamiltonian obviously breaks time reversal symmetry because the energy at k and $-k$ are not the same. In fact, individually, P and T symmetries cannot constrain the Hamiltonian to be real and symmetric because these symmetries relate $H(k)$ and $H(-k)$. On the other hand, PT symmetry relates $H(k)$ with $H(k)$, because P and T individually relates $H(k)$ and $H(-k)$. This is because individually, P and T acts nonlocally in the momentum space, but their combination acts locally in the momentum space. In fact, it is known that when $P^2 = 1$ and $T^2 = 1$, it is possible to choose a basis in which $PT=K$, where K is the complex conjugation. Thus, in this basis, the constraint is $H(\mathbf{k}) = PT H(\mathbf{k})(PT)^\dagger = KH(\mathbf{k})K = H^*(\mathbf{k})$.

This is the reason that the PT symmetry constrains the Hamiltonian to be real, which is a condition that is necessary to define the topological charge q (in addition to $H(\mathbf{k})$ being a 3-by-3 matrix).

In the revised manuscript, we have added an explanation regarding the role of the PT symmetry in Section I of the main text.

Question 4: The authors investigated how TATP charge q constrains the nodal structure of symmetry-protected TATPs when we perturb the Hamiltonian so that the relevant symmetry is relaxed, while the conditions required to define q are maintained. While some interesting consequences are observed due to TATP, there is no connection between the topological features of the resultant nodal structure (such as some topological invariants depicting the linkage of the nodal lines) and the TATP charge q . Based on the authors’ reasoning, this connection seems to be possible but was not done.

Authors’ response: We thank the reviewer for taking an interest in the linking structure resulting from the TATP. Actually, we have given the exact relation between the topological charge, the linking structure, and its proof in the Supplementary Information V. Because it is technical and lengthy, we have not included it in the main text. Instead, we have commented in Sec. IV that a further explanation for the linking structure is given in the Methods (‘‘Linked nodal structure protected by q ’’), where we explain why there should be a linking structure. The essence of the argument given in the Methods is that the black nodal ring can be contracted and gapped out if it is not threaded by the red nodal lines as in Fig. 3d. However, such a situation is not compatible with the skyrmion charge of the uppermost energy band, which guarantees that the uppermost band cannot be gapped out from the lower two bands (these gap closing points form the black nodal ring). The exact relation between the topological charge and the linking structure and its proof is given in the Supplementary Information (this was mentioned in the Methods section): on a surface having the black nodal ring as its boundary, the total vorticity of the 2D Dirac points (formed by intersection of the surface with the red nodal lines) is given by $2\pi_{sk}$.

Because the Supplementary Information is lengthy, we have specified exactly which sections are being referenced in our revised manuscript.

Question 5: In section 5 “AVOIDING THE DOUBLING THEOREM”, the authors discussed some examples of how this is achieved. However, the authors did not really establish theoretically why this well-known theorem is not relevant to TATPs --exactly what part of the assumption of this theorem is violated? Given many qualitative analyses in the manuscript, the lack of a solid reason to account for the avoiding of the doubling theorem is a bit disappointing to this reviewer.

Authors’ response: We thank the reviewer for asking this important question. In the manuscript, we explained two ways to avoid the doubling theorem, in the CsCl lattice and in the 3D Lieb lattice. To understand why the doubling theorem is avoided, let us first recall the argument that is used to show the doubling theorem (see Sec. V). We first assume that the integer valued topological charge can be defined for a two-dimensional (2D) slice in the BZ. In the case of the doubling of the TATP, the relevant integer valued topological charge is the Euler number computed for the two bands corresponding to the T modes of a TATP. Since the Euler number changes by ± 2 across the TATP, the periodicity of the BZ (see Fig. 4d) can be satisfied only if there is another TATP with Euler number ∓ 2 . However, this argument can be avoided when it is not possible to define the Euler number for a 2D slice in the BZ. In our manuscript, we gave two representative ways that this can happen.

Let us first consider the CsCl lattice. To understand why the doubling theorem is avoided, let us recall that the topological charge q that we are considering requires that around the triple point, there is a gap between two of the energy bands and the remaining energy band. For a TATP, the topological charge of the two energy bands is the Euler number, which can be defined regardless of the total number of energy bands in the system (as we have explained in Secs. I and II). Because the Euler number is a \mathbb{Z} valued quantity, we might expect that nodal structure with non-zero Euler number (i.e. the TATP) will not exist alone, as explained above.

However, in the case of CsCl, the doubling theorem is avoided: for the three lowest energy bands (let us call them the acoustic phonons), the only TATP is at Γ . To explain this, let us note that there is an energy gap between the lowest two acoustic phonons (the T modes) and the highest acoustic phonon (the L mode) around the TATP. However, this gap is not maintained throughout the BZ. In fact, as we have explained in Sec. V, there is a degeneracy between the L and one of the T modes along the RM line, as can be seen in Fig. 2b. This means that we cannot choose a 2D plane in the BZ as in Fig. 4d on which there is an energy gap between the L and the T modes throughout the 2D plane. Since such a gap must be satisfied to define the Euler number in a 2D plane, we see that it is not possible to define the Euler number for the two lowest energy bands on the 2D slice of the BZ when there is a band degeneracy between the L and T modes. This is why the periodicity argument that we have reviewed above cannot be applied to the TATP in CsCl.

In contrast to the phonon energy spectrum of CsCl, the energy spectrum in the 3D Lieb lattice does allow such a partitioning between the T modes and the L mode throughout the BZ. Here, as before, the T modes refer to the two modes with Euler number when computed on a sphere surrounding the TATP, and the L mode refers to the mode with skyrmion number on the same sphere. Therefore, one may expect that when we consider two parallel 2D planes in the BZ on the opposite sides of the TATP as in Fig. 4d, the difference between the Euler numbers computed on the two planes should be 2, and there should be a doubling of TATP. However, the argument for the doubling fails because of the Zak phase: when the Zak phase along any direction in the 2D plane is π , it is not possible to define the Euler number in the 2D plane at all. This is because two T modes do not form an orientable vector bundle when the Zak phase is π , whereas Euler number is defined only for orientable bundles, as we have explained in Sec. V.

The two examples discussed here are two representative ways to avoid the doubling theorem. In both examples, the doubling is avoided because it is not possible to define the Euler number on a 2D slice in the BZ. In the CsCl lattice, the Euler number cannot be defined because it is not possible to partition the energy bands on the 2D slice in the BZ into the L and the T modes. In the 3D Lieb lattice, the Euler number cannot be defined on the 2D slice in the BZ because the Zak phases are nontrivial.

In the revised manuscript, we have added some further explanation in Sec. V for clarity.

In summary, we think that the main reason the reviewer feels that the analysis was qualitative is due to the confusion caused by the fact that we have not clarified the relation between TATP defined in strictly three-band systems and the TATP that can appear in systems with more than three bands. Another possible reason that the reviewer feels that our analysis is qualitative seems to be caused by the fact that we have not pointed to the exact section in the Supplementary Information that are being referenced in the main text for the proof (e.g. the proof for the linking structure). We believe that these are the reasons that the reviewer felt that our work was not carried out in depth. In the revised manuscript, we have clarified the possibility of confusion about the three-

band conditions, and also clarified exactly which section in the Supplementary Information is being referenced in the main text. To further eliminate sources of confusion, we have added a section titled “Note on terminology” in Methods. As we have clarified in this response, we feel that we have given a solid reasoning for all of the statements we have made in our manuscript. Because all the concerns raised by the reviewer are appropriately addressed in our response and the revised manuscript, we hope that the reviewer will have a better opinion on our revised manuscript.

Reviewer #3 (Remarks to the Author):

In this paper, Park et al have established the topological nature of a special "node" in the band of phonons, which is the zero-momentum-zero-energy nexus of the triplet Goldstone modes. It is shown that a pair of topological invariants, the Euler number e and the Skyrmion number q , describe the above topology. Similar discussion applies to triplet node at higher energy protected by cubic symmetries. Possible consequences for this type of topology are discussed briefly.

For someone who is familiar with Ref.[35] from the same group, and an earlier paper from another group which was not cited, the above result adds limited information to the existing literature. The work at hand is more or less an application of the existing theory to the acoustic phonons. For that reason, I think the work is not seminal enough as a theoretical breakthrough. On the other hand, I appreciate the idea that a rather abstract \mathbb{Z}_2 topological invariant (which is the Euler number mod 2) has found its realization in one of the most common phenomenon of acoustic waves that appear in almost every solid. Such identification, ideally, should come with new physical effect to be experimentally observed, yet the discussion of possible effects originating from the new topology is too brief to be convincing or helpful for experimentalists. For that reason, I think the work has not enough potential to drive new experiments. Based on the above observations, I conclude that the physics shown in the current form is not sufficiently interesting for publication in Nature Communications.

Authors' response: We thank the reviewer for appreciating the fact that the topological charge q can appear in the common phenomenon of acoustic waves, which was the main conclusion we wanted to convey. Because q is a delicate topological charge, which is often looked on as something artificial due to the constraint on the number of energy bands required to define the charge, we think that this result can be of wide interest. Based on the reviewer's comments, we believe that the main reason the reviewer does not believe that our work is suitable for publication in Nature Communications is due to the overlap with Ref. [35]. However, as we explain below, the topological charge we studied in the present manuscript cannot be analyzed by a simple application of the theory in Ref. [35]. Actually, the reference to Ref. [35] in the present manuscript was not in regards to the \mathbb{Z}_2 monopole nodal lines. In fact, \mathbb{Z}_2 monopole nodal lines are not directly relevant to this work as we explain below. This is also the reason that other works on \mathbb{Z}_2 monopole nodal lines were not cited. We hope that once this misunderstanding is resolved, the reviewer will have a better opinion of our manuscript.

Sec.I and Sec.II:

In earlier works, it has been shown that $\pi_2(O(M+N)/O(M)*O(N))=\mathbb{Z}_2$ for $M, N>2$, and also that for $N=2$ (or $M=2$), this invariant becomes a \mathbb{Z} -invariant (later named as the Euler number). Here the authors are dealing with $M=1, N=2$, so a straightforward calculation would show that $e=2$, or $z=1$.

Authors' response: We thank the reviewer for this comment, since this point could also have caused other readers to feel that the topological charge we have studied is a simple application of existing theories such as Ref. [35]. As pointed out by the reviewer, the topological invariants for such cases have been computed in previous works, such as Ref. [34]. However, we are not aware of any work studying triple points with the charge $\pi_2(O(3)/O(1) \times O(2))$. Furthermore, the charge $q = (\pi_{sk}, e) = (1,2)$ does not reduce to the charge $\mathbb{Z}_2 = 1$ (it reduces to $\mathbb{Z}_2 = 0$ as e is even) when additional bands are present, if this is what the reviewer is implying by the comment "a straightforward calculation would show that $e=2$, or $z=1$." For this reason, properties of nodal structures with $\mathbb{Z}_2 = 1$ does not apply to the topological charge q .

Sec.III:

This section is good. It gives clear criterion when the topological invariant becomes nontrivial in terms of the elastic constants, and relates this criterion to the stability criterion. One cannot derive such information trivially from existing results.

It is interesting to see, however, that some phonon bands carry $q=0$. Is it possible to have $q>1$? If so I would find it very interesting.

Authors' response: We thank the reviewer for asking this interesting question. We do not think that it is possible for acoustic phonons to have $\pi_{sk} > 1$ (we take it that the reviewer is referring to the skyrmion charge by q) as we explain below. For simplicity, let us assume a cylindrical symmetry about the z axis. Also, let us note that in the isotropic case, the dynamical matrix is given by

$$H = v_T^2 k^2 \mathbf{1}_{3 \times 3} + k^2 (v_L^2 - v_T^2) \begin{pmatrix} \sin^2 \theta \cos^2 \phi & \sin^2 \theta \cos \phi \sin \phi & \sin \theta \cos \theta \cos \phi \\ \sin^2 \theta \cos \phi \sin \phi & \sin^2 \theta \sin^2 \phi & \sin \theta \sin \phi \cos \theta \\ \sin \theta \cos \theta \cos \phi & \sin \theta \sin \phi \cos \theta & \cos^2 \theta \end{pmatrix}$$

Noting that $\pi_{sk} > 1$ requires that the L mode be a map $S^2 \rightarrow S^2$ with degree greater than one, we can obtain such wavefunction texture by replacing $\phi \rightarrow 2\phi$. However, such a dynamical matrix is not a smooth function in k . For example, the term $k^2 \sin^2 \theta \sin^2 2\phi = 4k^2 \sin^2 \theta \sin^2 \phi \cos^2 \phi = 4k_y^2 \cos^2 \phi = 4k_y^2 \times \frac{k_x^2}{k_x^2 + k_y^2}$ is not smooth in k . This can be rectified only by introducing terms that are quartic in momentum, which is not compatible with elastic continuum theory, which yields only dynamical matrix that are quadratic in momentum.

Sec.IV:

In Ref.[35], authors from the same group give very comprehensive discussion of what happens as the nodal point breaks into a nodal line, and the linking structure that is basically the same as Fig.3d. Rather, in Ref.[35], the discussion is even more general than here, because they were dealing with arbitrary M and N. Simply applying the general results to M=1, N=2, one immediately finds the linking structure.

A related question is what happens if $q > 1$ ($e > 2$)? Should one expect a more complicated linking structure than Fig.3d?

Authors' response: We thank the reviewer for this comment, which can be a source of confusion. As the reviewer points out, based on the observation that the black nodal ring in our manuscript and the Z2 monopole nodal lines in Ref. [35] both show some form of linking structure, it can appear that the two are of the same origin. However, the linking structure in our manuscript cannot be explained by simply applying the argument in Ref. [35] as we now explain.

Let us recall that in Ref. [35], the authors consider nodal lines (forming a ring) with the Z2 monopole charge (computed on a sphere surrounding the nodal line), and show that it should be linked with another nodal line due to this charge. To show this, the authors

“continuously deform the sphere wrapping a NL γ , by gluing the north and south poles at the center, into a thin torus completely enclosing γ . As long as the band gap remains finite during the deformation, w_2 is invariant since the gluing of the north and south poles does not create a monopole... We assume that the torus is thin enough so that all occupied bands on it are nondegenerate. In this limit, according to the Whitney sum formula, w_2 satisfies the following relations modulo two ...”

First, let us note that this argument uses the Whitney sum formula for the second Stiefel-Whitney class for a nodal line with $w_2=1$. However, we are considering the case when $e = 2$ for the two lower energy bands in Fig. 3d in our manuscript (computed on a sphere surrounding the black nodal ring), which implies that $w_2=0 \pmod{2}$. Since the nodal structure considered in our manuscript has $w_2=0$, the argument in Ref. 35 does not carry over, and we require a separate proof for the three-band system.

Moreover, in a three-band system, the ‘smallest’ nontrivial topological charge is $q = (1,2)$, meaning that the smallest nonzero Euler number possible is 2. In other words, in a three-band system, we cannot even obtain a Z2 monopole nodal lines having $e = 1$ (corresponding to $w_2=1$). This implies that it is not even possible to regard the black nodal ring in our manuscript as a simple superposition of two Z2 monopole nodal lines. In fact, as pointed out in Ref. [35], the Z2 monopole nodal line requires a minimum of four bands, and the three-band case discussed in our manuscript is not a trivial application of the results in Ref. [35].

Next, let us answer the reviewer’s question on the case with $\pi_{sk} > 1$ (we take it that this is what the reviewer means by $q > 1$). Because the nodal structure in Fig. 3d with $q = (1,2)$ is the building block for the charge q , when $q = (2,4)$, we can simply regard it as two black nodal ring, each threaded by at least two nodal lines (in the terminology of Fig. 3d). If the black nodal rings merge, it will be threaded by at least four red nodal lines. Let us note that the total vorticity of the 2D Dirac points on a surface having the black nodal line as its boundary is twice the skyrmion number of the charge computed on the sphere surrounding it.

Sec.V:

This section is interesting. It involves the difference between the definition of topological invariants on S^n and that on T^n . I suggest to expand this section, specially where the authors claim that such an extension to T^n is possible if and only if the non-contractible loop has trivial Berry phase.

Authors' response: We thank the reviewer for this comment. Although the important quantity in this section was the Euler number (which is not the same as q as we have defined it), the reviewer rightly points out that the transition in the discussion of the charge defined on S^2 to T^2 may not have been obvious. It is of course also true that for a three-band system, q can be defined on a T^2 if and only if non-contractible loop has trivial Berry phase and there is a gap between two of the bands and the remaining band in a three-band system. This is actually quite easy to see because the Euler number can be defined only when the Berry phase is zero along all non-contractible loops in T^2 . Because the only obstruction to contracting the loops in T^2 is $\pi_1(O(3)/O(1) \times O(2)) = \mathbb{Z}_2$, which is the Berry phase, when the Berry phases are trivial, the topological charge on T^2 is the same as the topological charge on S^2 , which is nothing but q .

In the revised manuscript, we have added some further discussion on the Euler number to Sec. V.

Sec.IV:

I suggest that the authors significantly expand the discussion on experimental consequences. For me, several points should be clarified. (i) Since the nodal point is related to PT-symmetry, which is not preserved on any open surface, can I say that there is no topologically protected surface states related to this type of node? (ii) Since phonons do not couple to gauge fields, how should one observe the phonon angular momentum Hall effect? (iii) Can we have a concrete calculation of the orbital Hall effect in an electronic material hosting TP at Fermi energy?

The bottom line is that since the topological part of the theory is not new enough, the authors may consider shifting the emphasis to potentially new observable effects.

Authors' response: We thank the reviewer for carefully reading our manuscript and considering ways to improve it. Let us answer the reviewers' comments point by point below.

(i) Although the reviewer may know this, we would first like to point out that we have discussed the surface modes in the Discussion section, and we have investigated the problem in more detail in the SM, where we conclude that the TATP is not directly related to surface modes which is suggestive due to presence of surface acoustic waves in elastic systems. However, this is not related to the fact that PT symmetry is broken on systems with open surface.

This is because even if the PT symmetry is broken on the surface, topological surface states are not forbidden. As an example, let us consider a two-dimensional insulator with $e = 1$ (for further discussion on this example, see Ref. [18] Phys. Rev. X, 021013 (2019)). If we introduce edges to this system so that it has a square shape, for example, the PT symmetry is broken on each of the edges (note that the 2D system as a whole still preserves the PT symmetry). It nevertheless shows corner states because a 2D insulator with $e = 1$ is a higher-order topological insulator.

(ii) The reviewer is correct in saying that the phonon angular momentum does not directly couple to gauge fields like the electromagnetic field. However, let us note that the phonon angular momentum Hall effect discussed in Ref. [50] is a response to the thermal gradient, which can couple to phonon angular momentum. Furthermore, the phonon angular momentum Hall current induces accumulation of the phonon angular momentum on the surface. For an ionic crystal, the phonon angular momentum on the surface can induce edge magnetization due to the well-known relation $\mu = \frac{eZ_{eff}}{2M}L^z$, where Z_{eff} is the Born effective charge. This edge magnetization does couple to the electromagnetic field (gauge field). This was discussed in detail in Ref. [50]. Because the PAMHE is not well known, we have made additional comments regarding this point in the Discussion section.

(iii) The orbital Hall effect arising from TATP is discussed in detail in Refs. [51,52] as we have pointed out in the Discussion section, although the authors in these papers did not recognize the relation between the texture of the wavefunctions they study and the topological charge q . The point we wanted to make in the present manuscript is that the texture that was already pointed out as a source of the PAMHE and the orbital Hall effects are actually characterized by the topological charge q .

Although we appreciate the reviewer's suggestion of studying the orbital Hall effect arising from TATP in more detail, we do not feel that it is a suitable subject for this particular manuscript, firstly because the orbital Hall effect is not the main topic of our manuscript, and secondly, because we believe that a study of PAMHE and orbital Hall effect that goes beyond Refs. [50,51,52] would be suitable for a separate research paper. This is especially so because although TATP can be a significant source of the OHE, it is not the only source of OHE (just as Weyl points are not the only sources of anomalous Hall effect), and even in the case of TATP, the energetics will influence the magnitude of the orbital Hall effect. Therefore, finding a real material that hosts just the TATP at the fermi surface and studying the orbital Hall effect would make our paper too long and even motley. However, for the reference for the reviewer, we have analytically computed $\Omega_{k,n}$ that appears in the expression for the orbital Hall effect for the isotropic Hamiltonian. The orbital Hall conductivity is given by the expression

$$\sigma_{OH} = \frac{e}{\hbar} \sum_{n \neq m} \int \frac{d^3k}{(2\pi)^3} (f_{mk} - f_{nk}) \Omega_{nmk}^{Lz}, \Omega_{nmk}^{Lz} = \hbar^2 \text{Im} \left(\frac{\langle u_{nk} | j_y^{Lz} | u_{mk} \rangle \langle u_{mk} | v_x | u_{nk} \rangle}{(E_{nk} - E_{mk} + i\eta)^2} \right)$$

For TATP with Hamiltonian $[H_k]_{\alpha\beta} = ak^2\delta_{\alpha\beta} + bk_{\alpha}k_{\beta}$,

$$\Omega_{LT_1k}^{Lz} = -\Omega_{T_1Lk}^{Lz} = \hbar \frac{k_y^2 k_z^2 b + 2k_y^2 \tilde{k}^2 (2a + b)}{2\tilde{k}^2 k^4 b}$$

$$\Omega_{LT_2k}^{Lz} = -\Omega_{T_2Lk}^{Lz} = \hbar \frac{k_x^2 k_z^2}{2\tilde{k}^2 k^4}$$

$$\Omega_{T_1 T_2 k}^{L_z} = -\Omega_{T_2 T_1 k}^{L_z} = 0,$$

where $\tilde{k}^2 = k_x^2 + k_y^2$. We would like to note that although a direct relation between TATP and orbital Hall effect is complicated by the location of fermi energy, the energy spectrum around the TATP, and the difficulty of isolating just the TATP, the phonon angular momentum Hall effect results almost exclusively from the TATP at low temperatures, as discussed in detail in Ref. [50] (note that the relation to the topological charge was also overlooked there).

We hope that our response has cleared up the misunderstanding that the theory in this manuscript is a trivial corollary of the theory in Ref. [35]. Because this was the main reason that the reviewer did not feel that our work is new, we would like to ask the reviewer to kindly re-evaluate our manuscript based on our response.

REVIEWER COMMENTS

Reviewer #2 (Remarks to the Author):

I have read authors' responses to questions from all three referees. The authors have done a beautiful job in addressing many concerns and I tend to believe that this work may now probably make it to Nature Communications. The extensive exchange between the authors and the reviewers is highly academic and hence professional.

However, this reviewer is still confused by the role of T-symmetry. In their reply to reviewer 2, they said the following:

"Note that by stating that the dynamical matrix is 'real-valued', we are implicitly assuming that the time reversal symmetry is not broken in the phonon spectrum. Under this assumption, the dynamical matrix in the continuum limit is real valued. Please refer to our response to the reviewer 1's question 2 about this point on time reversal symmetry in elastic continuum limit."

This is precisely this reviewer's point in his first report. If T-symmetry is already assumed to be there, why should the authors keep emphasizing PT symmetry? That is, if T-symmetry is implicitly assumed from the very beginning, then only P-symmetry is relevant, not the PT symmetry. PT-symmetry is more relevant only if both P and T symmetries are individually broken.

In their response following the above statement, the authors then contradicted themselves by saying (in reply to Question 3 of reviewer 2) that the joint PT symmetry (not T-symmetry) can guarantee that the dynamical symmetry is real.

I hope that the above-mentioned self-contradiction is merely caused by authors' writing, not on the fundamental level.

Response to the reviewer

Reviewer #2 (Remarks to the Author):

I have read authors' responses to questions from all three referees. The authors have done a beautiful job in addressing many concerns and I tend to believe that this work may now probably make it to Nature Communications. The extensive exchange between the authors and the reviewers is highly academic and hence professional.

Authors' response:

We thank the reviewer for the positive evaluation of the revision of our manuscript.

However, this reviewer is still confused by the role of T-symmetry. In their reply to reviewer 2, they said the following:

"Note that by stating that the dynamical matrix is 'real-valued', we are implicitly assuming that the time reversal symmetry is not broken in the phonon spectrum. Under this assumption, the dynamical matrix in the continuum limit is real valued. Please refer to our response to the reviewer 1's question 2 about this point on time reversal symmetry in elastic continuum limit."

This is precisely this reviewer's point in his first report. If T-symmetry is already assumed to be there, why should the authors keep emphasizing PT symmetry? That is, if T-symmetry is implicitly assumed from the very beginning, then only P-symmetry is relevant, not the PT symmetry. PT-symmetry is more relevant only if both P and T symmetries are individually broken.

In their response following the above statement, the authors then contradicted themselves by saying (in reply to Question 3 of reviewer 2) that the joint PT symmetry (not T-symmetry) can guarantee that the dynamical symmetry is real.

I hope that the above-mentioned self-contradiction is merely caused by authors' writing, not on the fundamental level.

Authors' response:

We thank the reviewer for carefully reading the previous response letter and clarifying the reviewer's intention for Question 3 in the previous correspondence. The reviewer is certainly correct in saying that once we assume that the time-reversal symmetry is present in the dynamical matrix, only the presence of the inversion symmetry needs to be emphasized. However, we opted to emphasize the role of the PT symmetry for the following reasons:

1. The main role of the PT symmetry is in constraining the Hamiltonian to take real values. As we have explained in the previous correspondence, this is because we can take $PT=K$ (complex conjugation). Because the reality condition is crucial in defining the topological charge q , the PT symmetry is necessary for us to define this topological charge. Although this point may already be clear to the reviewer, let us again emphasize that *neither* P nor T by itself can constrain the Hamiltonian to be real, and it is the combined PT symmetry that constrains the Hamiltonian to be real. Furthermore, the condition that PT symmetry is present is more general than the condition that both P and T symmetries are present. For these reasons, we believe that even in the case when both the P and the T symmetries are present, it is better to emphasize the presence of the PT symmetry rather than the presence of both the P and the T symmetries when discussing the topological charge q of a real Hamiltonian. Please refer to the following comments in Sec. I "To define q , we require that H_k be 3×3 real symmetric matrix" and "The condition that H_k be real is satisfied if there is PT symmetry...".
2. In our manuscript, we also extended the discussion of the topological charge q to electronic bands. In electronic systems, the effect of time-reversal breaking is more apparent than in

phonons. Therefore, in electronic systems, there is no reason to limit the discussion to time-reversal symmetric systems, in contrast to phonons, where the appearance of time-reversal breaking is rather special. At this point, the reviewer might ask the following question: since we state that the TATP can appear in high symmetric points in the presence of O_h and T_h groups (in electronic systems), are we not assuming that both the P and the T symmetries are present even in electronic systems (since O_h and T_h is P-symmetric and we also assume PT-symmetry)? While this is a valid point, we still think that it is better to emphasize the PT symmetry, because in our manuscript, we also discuss the fate of the TATP under the perturbations that keep the Hamiltonian to be real. Such perturbations are allowed to break both the P and the T symmetries, but they must preserve the PT symmetry in order to observe the linking structure protected by the charge q (note that the Hamiltonian must be real in order to define q , but it does not have to be symmetric under both T and P). Thus, for this discussion, it is necessary to emphasize the PT symmetry.

3. Even for phonons, it is in principle possible for the P and the T to be broken, but for the PT symmetry to be present. Actually, in the previous response, our comment “we are implicitly assuming that the time reversal symmetry is not broken in the phonon spectrum” was made with a specific situation in mind. Namely, we were considering specific examples of how the time-reversal symmetric phonon Hamiltonian is modified in response to external magnetic field (manifested by the Mead-Truhlar term and Raman coupling). However, it is also possible for antiferromagnetic ordering to break the T symmetry and modify the phonon spectrum. Furthermore, antiferromagnetic ordering can be compatible with the PT symmetry, but it is not compatible with the time-reversal symmetry, so that in such cases, it would be the PT symmetry that constrains the phonon Hamiltonian to be real. Although time reversal symmetry breaking effects in phonon spectrum are usually small, we prefer not to completely exclude such possibilities. To summarize, it is true that in time-reversal symmetric crystals, the reality of the phonon Hamiltonian additionally requires only the inversion symmetry. However, in time-reversal broken crystals, the reality of the phonon (and electronic) Hamiltonian depends on the PT symmetry.

For the convenience of the reviewer, we provide a summary of relevant facts:

- The PT symmetry is necessary for constraining the Hamiltonian to be real.
- The Hamiltonian must be real in order to define the topological charge q .
- Neither P nor the T symmetry can individually constrain the Hamiltonian to be real.
- The Hamiltonian for acoustic phonons in the presence of time-reversal symmetry is real due to the presence of the (emergent) inversion symmetry (cf. response to Question 2 of reviewer 1 in the previous correspondence).
- The Hamiltonian for acoustic phonons does not have to be real in the presence of magnetic fields/magnetic order.
- However, even if the P and the T symmetry are broken, if the PT symmetry is present, the Hamiltonian is real. This can happen in antiferromagnetic systems.
- For the presence of triple degeneracy (TATP), we considered only the cases when both P and T symmetries are present.
- However, definition of q does not require both the P and the T symmetries, but only the PT symmetry. Thus, perturbations that break the triple degeneracy while maintaining the condition necessary to define q can break both P and the T symmetries, but not the PT symmetry. Let us note that this point was discussed in Sec. IV (“the definition of q only requires that the Hamiltonian be a 3×3 real symmetric matrix with a spectral gap between (the) L and the T modes, while the symmetry-protected TATP requires further constraints such as the O_h symmetry”: note here that the reality condition is equivalent to the presence of PT symmetry)

We hope that the above explanation clarifies the confusion.

To emphasize the role played by the PT symmetry for the perturbation that breaks the triple degeneracy, we added the following comment in Sec. IV:

“Let us note that we allow the possibility for the perturbation to break the P and the T symmetries, but the perturbation must preserve the combined PT symmetry to keep the Hamiltonian real.”

REVIEWERS' COMMENTS

Reviewer #2 (Remarks to the Author):

The authors have answered my question with great efforts. Delaying the review process is the last thing I hope to do, but some further discussion should be healthy. What the authors said did not explain their logically contradicting sentence below: (authors did not get the key point in my earlier report) :

Authors say: "Note that by stating that the dynamical matrix is 'real-valued', we are implicitly assuming that the time reversal symmetry is not broken in the phonon spectrum". According to this logic, if the matrix is real, then there is assumed time reversal symmetry. On the other hand, the authors keep highlighting that real matrix is associated with PT symmetry, rather than the T symmetry alone. So what does the authors really wish to say, by saying "implicitly assuming....". Can the authors fix this?

Reviewer #3 (Remarks to the Author):

I personally apologize to the authors and the editor for my overdue report.

I was previously convinced that, despite the authors' explanation in their reply, that Euler# = 2 corresponds to the $Z_2 = 1$ case. I was of the intention to use an explicit example to show this to the authors, until today when I found that my example proved exactly the opposite, that $E\# = 2$ corresponds to $Z_2 = 0$.

With the main misunderstanding resolved, I am convinced of the authors' major claim that TATP is a simple but novel example for realizing the $E\# = 2$ topological node.

I am fine with the rest of the reply and revision, except for the one on the avoidance of the NN-theorem. In fact, I prefer what the authors say in the reply to what they write in the paper, explaining well that the NN-theorem only works for S^n , but may be extended to T^n given trivial invariants for all non-contractible loops.

I recommend publication after the optional minor revision (mentioned above).

Reviewer #2 (Remarks to the Author):

The authors have answered my question with great efforts. Delaying the review process is the last thing I hope to do, but some further discussion should be healthy. What the authors said did not explain their logically contradicting sentence below: (authors did not get the key point in my earlier report):

Authors say: "Note that by stating that the dynamical matrix is 'real-valued', we are implicitly assuming that the time reversal symmetry is not broken in the phonon spectrum". According to this logic, if the matrix is real, then there is assumed time reversal symmetry. On the other hand, the authors keep highlighting that real matrix is associated with PT symmetry, rather than the T symmetry alone. So what does the authors really wish to say, by saying "implicitly assuming....". Can the authors fix this?

Authors' response:

We agree that the expression "Note that by stating that the dynamical matrix is 'real-valued', we are implicitly assuming that the time reversal symmetry is not broken in the phonon spectrum" is misleading. The correct statement should be "**Note that the dynamical matrix is real-valued when the system has PT symmetry. As P symmetry is restored in the elastic continuum limit near the triple degeneracy point even in noncentrosymmetric crystals, when the system has T symmetry, the dynamical matrix becomes real in both centrosymmetric and non-centrosymmetric crystals**". The meaning of "implicitly assuming" was that we focused on the time-reversal symmetric crystals.

Please find more detailed context of such an explanation, including the effect of time-reversal symmetry breaking, in our previous response. Let us stress that the above confusing expression, correctly pointed out by the reviewer, appeared only in our response letter in the first round of the review process. Since this expression is not directly related to the main point of our paper, there is no logical flaw in our paper. We hope that this resolves the remaining issue pointed out by the reviewer.

Reviewer #3 (Remarks to the Author):

I personally apologize to the authors and the editor for my overdue report.

I was previously convinced that, despite the authors' explanation in their reply, that Euler# = 2 corresponds to the $Z_2 = 1$ case. I was of the intention to use an explicit example to show this to the authors, until today when I found that my example proved exactly the opposite, that $E\# = 2$ corresponds to $Z_2 = 0$.

With the main misunderstanding resolved, I am convinced of the authors' major claim that TATP is a simple but novel example for realizing the $E\# = 2$ topological node.

I am fine with the rest of the reply and revision, except for the one on the avoidance of the NN-theorem. In fact, I prefer what the authors say in the reply to what they write in the paper, explaining well that the NN-theorem only works for S^n , but may be extended to T^n given trivial invariants for all non-contractible loops.

I recommend publication after the optional minor revision (mentioned above).

Authors' response:

We thank the reviewer for carefully reviewing our manuscript, and recommending our manuscript for publication at Nature Communications. We fully understand the difficulty that the

reviewer had about the topological charge, as the point on the topological charge can be subtle, especially because both topological charges are associated with linking structures.

In regards to the NN theorem, we appreciate the reviewer for making suggestions to improve our manuscript. Here, we believe that the reviewer is referring to our response to the following comment in the first round of review:

“Sec.V:

This section is interesting. It involves the difference between the definition of topological invariants on S^n and that on T^n . I suggest to expand this section, specially where the authors claim that such an extension to T^n is possible if and only if the non-contractible loop has trivial Berry phase.”

Our reply was that the possible topological charge on T^2 is the same as the possible topological charge on S^2 when there is no topological obstruction to shrinking the non-contractible loops on T^2 .

In the previous discussion we mentioned above, the main point is regarding the ‘difference’ between the possible topological charges on T^2 as opposed to S^2 . Because the Reviewer is now referring to the status of NN theorem on S^3 vs T^3 , we are slightly confused on whether the Reviewer is referring to the topological charge on 2D as mentioned above, or the NN theorem in 3D, since they are two different topics and we did not explicitly mention the NN theorem in our previous response. Our understanding is that by the statement “the NN-theorem only works for S^n , but may be extended to T^n given trivial invariants for all non-contractible loops”, the reviewer means that on S^n , there is no orientability obstruction to defining the Euler number (the ‘difference’ between the topological charge on S^2 and T^2), so that there should be a doubling of TATP on for S^3 . Then, we can reverse the process of the deformation involved in $T^3 \rightarrow T^3/(S^1 \vee S^1 \vee S^1) \approx S^3$ (here, \vee is the wedge sum), to obtain the NN theorem on the torus from that on the sphere. While this is true, we think that this could be confusing to the reader for a few reasons.

First, the NN theorem is usually discussed on the torus by using the periodicity argument (this is how we proceeded in Sec. V), so that stating the NN theorem on S^n is slightly awkward (the first part of the logic in $\langle \text{NN holds on } S^3 \rightarrow \text{extension to } T^3 \rangle$).

Second, to follow the logical flow $\langle \text{NN holds on } S^3 \rightarrow \text{extension to } T^3 \rangle$, we need to first explain why doubling holds on S^3 . Although it is possible to directly show this, and extend to T^3 by reversing $T^3 \rightarrow T^3/(S^1 \vee S^1 \vee S^1) \approx S^3$, we think that this takes us too far away from the core discussion in this section. This is because the main message we wanted to get across in this section is that the doubling of TATP can be avoided due to the two reasons we have explained in the manuscript, although the viewpoint suggested by the Reviewer is certainly interesting.

Third, we do not use the doubling of the TATP on S^3 anywhere in our manuscript, so we could not find a suitable way to fit it naturally into the flow of our manuscript without disrupting the flow.

Since the Reviewer kindly suggested this as an optional revision, we have decided to keep the logical flow of Sec. V (“Avoiding the doubling theorem”) as is.